# Full-length single-cell BCR sequencing paired with RNA sequencing reveals convergent responses to pneumococcal vaccination

Duncan M. Morgan [1,2], Yiming J. Zhang [1,3], Jin-Hwan Kim[4], MaryAnn Murillo[4], Suddham Singh[4], Jakob Loschko[4,8], Naveen Surendran[4], Ognjen Sekulovic[4], Ellie Feng [1,3], Shuting Shi[1,2], Darrell J. Irvine [1,3,5,6], Sarita U. Patil[7], Isis Kanevsky[4], Laurent Chorro[4,9] & J. Christopher Love [1,2] ✉

Single-cell RNA sequencing (scRNA-seq) can resolve transcriptional features from individual cells, but scRNA-seq techniques capable of resolving the variable regions of B cell receptors (BCRs) remain limited, especially from widely-used 3′-barcoded libraries. Here, we report a method that can recover paired, full-length variable region sequences of BCRs from 3′-barcoded scRNA-seq libraries. We first verify this method (B3E-seq) can produce accurate, full-length BCR sequences. We then apply this method to profile B cell responses elicited against the capsular polysaccharide of *Streptococcus pneumoniae* serotype 3 (ST3) by glycoconjugate vaccines in five infant rhesus macaques. We identify BCR features associated with specificity for the ST3 antigen which are present in multiple vaccinated monkeys, indicating a convergent response to vaccination. These results demonstrate the utility of our method to resolve key features of the B cell repertoire and profile antigen-specific responses elicited by vaccination.

B cells play a crucial role in the adaptive immune response by producing antibodies against various pathogens. The diversity of this response starts with genetic recombination and diversification of the B cell receptor (BCR)[1]. Upon antigen recognition, naïve B cells can undergo clonal expansion, somatic hypermutation (SHM), and class-switch recombination, further diversifying the BCR repertoire[2–4]. B cells that enter germinal centers can undergo differentiation into either plasma cells or memory B cells, leading to the acceleration of the humoral response and the establishment of immunological memory[5–7].

Previous analyses of the BCR repertoire have provided insights into antigen-specific immune responses in many contexts, including auto-immune diseases, allergies, and vaccination[8–14]. Commonly, the variable regions of the BCR are analyzed en masse using next-generation sequencing (NGS)[8–14]. While this approach provides substantial throughput, it fails to acquire naturally paired heavy and light chain sequences, which are required to express and functionally interrogate the specificity, affinity, and clonality

of the antibodies. Most frequently, to obtain paired heavy and light chains, single B cells are sorted into microliter plates, and BCRs are subsequently amplified with nested PCR followed by Sanger sequencing, limiting the overall throughput[15–18]. More recently, strategies based on NGS have enabled the acquisition of pairings of heavy/light chains and simultaneous readouts of BCR specificity[19–22], but these methods remain limited in their ability to simultaneously assess the transcriptomes of single B cells. In addition, these methods remain difficult to apply to sparse samples, such as small populations of antigen-specific cells isolated from the blood or biopsy samples available from human patients.

Single-cell RNA sequencing (scRNA-seq) enables the analysis of whole transcriptomes of single cells with substantial throughput and has revolutionized studies of gene expression[23,24]. Pioneering studies combining the analysis of the BCR repertoire with B cell phenotypes have furthered our knowledge of how BCRs relate to the fates and functions of B cells[25–28]. Despite this, many scRNA-seq platforms remain limited in their ability to

¹Koch Institute for Integrative Cancer Research, MIT, Cambridge, MA, USA. ²Department of Chemical Engineering, MIT, Cambridge, MA, USA. ³Department of Biological Engineering, MIT, Cambridge, MA, USA. ⁴Vaccine Research and Development Pfizer Inc. Pearl River, New York, NY, USA. ⁵Howard Hughes Medical Institute, Chevy Chase, MD, USA. ⁶Department of Materials Science and Engineering, MIT, Cambridge, MA, USA. ⁷Center for Immunology and Inflammatory Diseases, Massachusetts General Hospital, Harvard Medical School, Boston, MA, USA. ⁸Present address: Deerfield Management, New York, NY, USA. ⁹Present address: Regeneron, Tarrytown, NY, USA. ✉e-mail: clove@mit.edu

obtain BCR variable region sequences. Early demonstrations of scRNA-seq based on the isolation of individual cells into microliter plates followed by full-length RNA sequencing enabled the in silico reconstruction of BCR variable region sequences[29,30]. By their nature, these solutions exhibit reduced throughput compared to massively parallel platforms for scRNA-seq, which utilize short sequence reads to obtain digital counts of gene expression, rather than full-length RNA sequence coverage. Because the BCR variable region is located on the 5′-end of the BCR transcript, sufficient BCR variable region coverage can often be obtained from massively parallel 5′-barcoded library constructions[31], such as the 10x Genomics' Single Cell Immune Profiling platform. However, the orientation of the BCR transcript poses a significant limitation for 3′-barcoded library constructions, which obtain minimal coverage of the BCR variable region in their native forms. Importantly, 3′-barcoded libraries constructed using either academic (Drop-Seq, Seq-Well, inDrop, Microwell-seq, SPLiT-seq)[32–37] or commercial (10x Genomics Single Cell 3′ Gene Expression (GEX)) platforms currently account for a substantial fraction of both previously published and newly-generated scRNA-seq data. Reported methods to enable the recovery of BCR variable region sequences from 3′-barcoded libraries have relied on specialized RNA capture reagents (DART-seq)[38] or the use of multiple sequencing modalities (i.e., sequencing-by-synthesis plus long-read nanopore sequencing)[39,40]. These constraints have limited the broad adoption of these techniques based on available resources, total costs, or both.

Here, we present an approach for the recovery and sequencing of paired, full-length BCR variable region sequences compatible with 3′-barcoded scRNA-seq libraries, including 10x Genomics 3′ GEX, Seq-Well, and other systems[32,35,41]. This method, which we call BCR repertoire from 3′ gene Expression sequencing (B3E-seq), is cost-effective, scalable to large numbers of samples, and useful for recovering BCR sequences from archived samples. We first established the ability of our approach to recover accurate BCR sequences from human peripheral blood mononuclear cells (PBMC). We then used the approach to profile both transcriptional and clonotypic features present among antigen-specific B cells elicited by protein-polysaccharide conjugate vaccines in rhesus macaques.

## Results

### Targeted recovery and sequencing of full-length, variable region BCR sequences

In 5′-barcoded scRNA-seq library constructions, amplicons containing the cellular barcode and BCR variable region are generated by the full-length whole-transcriptome amplification (WTA) and subsequent amplification of BCR transcripts using primers specific for the universal primer site (UPS) and the BCR constant region (Supplementary Fig. 1a, b). The resulting amplicons typically undergo random fragmentation and further amplification to generate sequencing libraries, in which the sequences of the cellular barcode and the BCR can be simultaneously determined (Supplementary Fig. 1c). In contrast, while full-length BCR transcripts are captured by 3′-barcoded libraries, the BCR variable region is located on the opposite end of the construct from the cellular barcode (Fig. 1a). The process of random fragmentation used to generate size-defined, whole-transcriptome libraries prevents direct readout of both the single-cell barcode and the full-length BCR variable region (Fig. 1a).

To address this constraint, we extended a method for recovering the complementarity-determining region 3 sequences of T cell receptors[42]. In our approach, a portion of the 3′-barcoded WTA product is used to enrich for BCR sequences by probe-based affinity capture using biotinylated oligonucleotides that target the constant regions of BCR heavy and light chain isotypes (Fig. 1b). This product is then reamplified using the same UPS as the original WTA reaction. The resulting BCR-enriched product is subsequently modified by primer extension using a set of oligonucleotides comprising a shared 5′ UPS (UPS2) linked to sequences specific for the leader (L) or framework 1 (FR1) region of BCR heavy and light chain variable (V) segments (Fig. 1c). The product of this primer extension step is finally amplified with primers containing sequencing platform-dependent adapters linked to regions specific for either UPS2 (5′-end of the construct) or the original UPS (3′-end of the construct) (Fig. 1d). The resulting amplicons can then be sequenced using two overlapping reads in opposite directions, using the UPS2 sequence appended during primer extension (5′ to 3′ direction) and custom sequencing primers targeting BCR constant regions (3′ to 5′ direction). In addition, a third read is used to obtain the cellular barcode and unique molecular identifier (UMI) appended during

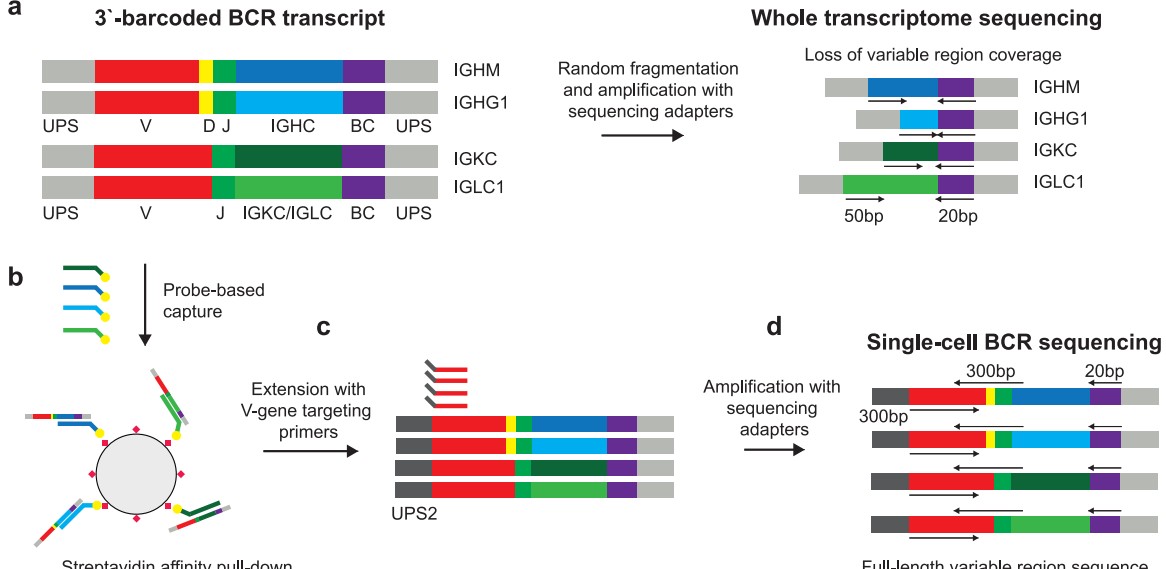

**Fig. 1 | Workflow of recovering BCR sequences from 3′-barcoded single-cell gene expression sequencing libraries. a** For 3′-barcoded scRNA-seq libraries, the random fragmentation leads to a substantial loss of coverage of the BCR variable region. **b** In our approach, BCR transcripts are enriched from cDNA products using probe-based affinity capture with oligonucleotides targeting the constant regions of BCR isotypes. **c** This enrichment product is then modified by primer extension with hybrid UPS2/V-gene targeting primers and amplified with sequencing adapters to produce a size-defined sequencing library containing the full length of the BCR variable region as well as the single-cell cellular barcode and UMI. **d** During sequencing, one read is used to capture the cellular barcode and UMI, and two overlapping reads in opposite directions are used to capture the full length of the BCR variable region, which can be assembled in silico and matched to single-cell transcriptomes using the corresponding cellular barcode.

library preparation. The custom BCR sequencing primers are designed to capture around 9 to 11 bases of the constant region, enabling differentiation of BCR constant region isotypes (e.g., IgM vs. IgG). If further determination of BCR subclass is desired (e.g., IgG1 vs. IgG2), this information can be obtained by analyzing the coverage of the BCR constant region obtained in whole-transcriptome (WT) sequencing, which is abundant in 3′-barcoded whole transcriptome library preparations[43–45], or additionally captured by protocols that stain cells with oligonucleotide-labeled antibodies, such as CITE-seq[46,47].

To process the BCR data generated using this approach, we developed a pipeline in which we first group individual sequence reads by molecular identity (unique combination of cellular barcode and UMI) and determine molecular consensus sequences for the both BCR sequence reads. We then assemble these reads in silico to reconstruct a full-length BCR sequence, spanning from the V-region primer binding site to the beginning of the constant region. We utilize a further processing step to establish the single-cell consensus of paired BCR sequences by collapsing the heavy and light chain molecular consensus sequences under each cellular barcode sequence.

## Recovery of BCR sequences from human peripheral blood mononuclear cells

To demonstrate that our method can recover BCR sequences, we performed an initial experiment in which we enriched B cells from human PBMC using magnetic associated cell separation and analyzed the resulting suspensions of cells with two 3′-barcoded scRNA-seq platforms: Seq-Well and 10x Genomics 3′ GEX v3 (Fig. 2a–f). In the resulting WT data, we annotated cells according to gene expression and identified two B cell populations, which correspond to naïve B cells and memory B cells (MBC), one small population of plasmablasts (PB), and non-B cells (Fig. 2a, d, Supplementary Fig. 2a, b).

We then recovered BCR sequences using our method (Fig. 2b, c, e, f). We obtained similar rates of the recovery of BCRs from Seq-Well libraries and 10x libraries; on average, from Seq-Well libraries, we obtained full-length heavy chain sequences from 66.7% of B cells, light chain sequences from 60.3% of B cells, and paired heavy and light chain sequences from 42.2% of B cells, and from 10x libraries, we recovered heavy chain sequences from 56.1% of B cells, light chain sequences from 89.9% of B cells, and paired chain sequences from 52.2% of B cells.

To assess the efficiency of our method, we determined the number of unique molecules mapping to the BCR heavy and light chains in whole transcriptome libraries and compared these counts to the corresponding number of unique molecules recovered from the same cells with our method. We found positive correlations between the number of molecules of heavy chain, kappa, and lambda transcripts in whole transcriptome libraries and the number of unique molecules recovered from our method for both Seq-Well and 10x libraries (Fig. 2g–i), suggesting that the quality of the input whole-transcriptome library influences the amount of BCR information that can be recovered. We also observed that the recovery of BCRs was dependent on the phenotype of the B cells, with elevated levels of recovery from PBs and moderately reduced rates of recovery from MBCs, consistent with differential levels of BCR transcript expression by these cells in the corresponding whole-transcriptome data (Fig. 2c, f, Supplementary Fig. 2c–f).

We also noted that our method appeared to recover BCR sequences from a minority of non-B cells in Seq-Well libraries (Fig. 2c). We further examined the single-cell expression profiles of these cells and found that most of these cells contained detectable mapping to BCR heavy and light chains in whole-transcriptome libraries (Supplementary Fig. 2g, h). Indeed, non-B cells with at least one count of heavy chain in WT libraries were 15.9 times more likely than other B cells to have a recovered heavy chain, and non-B cells with at least one count of lambda or kappa chain were 20.3 times more likely than other B cells to have a recovered light chain. This finding suggests that the majority of these likely artifactual sequences were not a product of our method for enrichment and BCR library preparation, but

rather were generated in the preceding steps of single-cell library preparation.

To confirm that our bioinformatic pipelines enabled the accurate reconstruction of BCR sequences, we compared individual UMI sequences recovered from the same cell to the cellular consensus BCR sequence determined by our pipeline. We found strong agreement between individual UMI sequences and cellular consensus sequences in both Seq-Well and 10x libraries, supporting a high degree of sequence accuracy (Supplementary Fig. 3a, b). A large portion of sequences that did not match the cellular consensus sequence demonstrated little sequence similarity to the cellular consensus sequence (>10 mismatched bases), suggesting that they result from multiple, unrelated BCR transcripts associated with one cell barcode. This phenomenon may result from physical cell doublets in single-cell droplets or microwells, contamination with ambient BCR sequences, or the generation of chimeric transcripts during PCR amplification. We also assessed the degree of sequencing saturation obtained in these BCR libraries and found that all samples exhibited a plateau in the number of BCR sequences recovered per additional sequencing read, suggesting that additional sequencing depth was unlikely to lead to the recovery of substantially more additional BCR sequences (Supplementary Fig. 3c–e). Together, these results support that our method enables the recovery of accurate, full-length BCR sequences from 3′-barcoded scRNA-seq libraries.

## Recovered BCR sequences demonstrate concordance with single-cell whole-transcriptome libraries

To further assess the accuracy of BCR sequences produced by our method, we examined their agreement with their corresponding single-cell transcriptomes. We first approximated the heavy chain isotypes of single-cell transcriptomes using reads mapping to the heavy chain constant region and compared these isotypes to those recovered using our method (Fig. 3a, b). We found that for cells expressing IgM or IgD isotypes in whole-transcriptome sequencing, we recovered a mixture of IgM and IgD BCRs. For cells expressing IgG and IgA sequences, we recovered predominantly either IgG or IgA BCRs, respectively (Fig. 3c, d). Additionally, we recovered mostly IgM and IgD BCRs from naive B cells and increasing frequencies of class-switched IgG and IgA BCRs from MBCs and PBs (Fig. 3e, f). These results demonstrate that our method faithfully preserved information related to BCR isotype.

We then analyzed the agreement of V-gene mappings between the whole transcriptome and BCR sequences recovered by our method. To avoid ambiguities related to differences in the genome and immunoglobulin gene references used in this study, we focused this analysis on a subset of V-genes that existed in both the human genome and the IMGT reference used in this study. We found clear trends of concordance between IGHV, IGKV, and IGLV gene usage present in whole-transcriptome libraries and the corresponding recovered BCR sequences, suggesting that our method also faithfully recovers information related to the variable region of the BCR (Fig. 3g, h, Supplementary Fig. 4a, b).

Lastly, we computed the somatic mutation frequency of recovered heavy and light chain BCR sequences (Fig. 3i, j). We found clear trends of increasing mutation frequency from naïve to MBC to PB cell phenotypes (Fig. 3k–n) and from IgM to IgG and IgA isotypes (Fig. 3o, p). Overall, these results indicate a concordance between the transcriptomes of B cells and their BCR sequences recovered with our method.

## Recovered BCR sequences demonstrate high levels of accuracy and sensitivity

To further assess the accuracy of the recovered sequences, we evaluated the ability to recover BCRs from a human embryonic kidney (HEK) cell line transiently transfected with separate plasmids encoding the heavy and light chains of a fully human IgG1/κ BCR (U6) (Supplementary Table 1). We diluted this transfected cell line into B cells isolated from an additional sample of PBMC and analyzed the resulting cell suspension with scRNA-seq. In addition to HEK cells, we identified clusters of non-B cells, naïve B cells, MBCs, and PBs in the resulting transcriptional data (Fig. 4a, b).

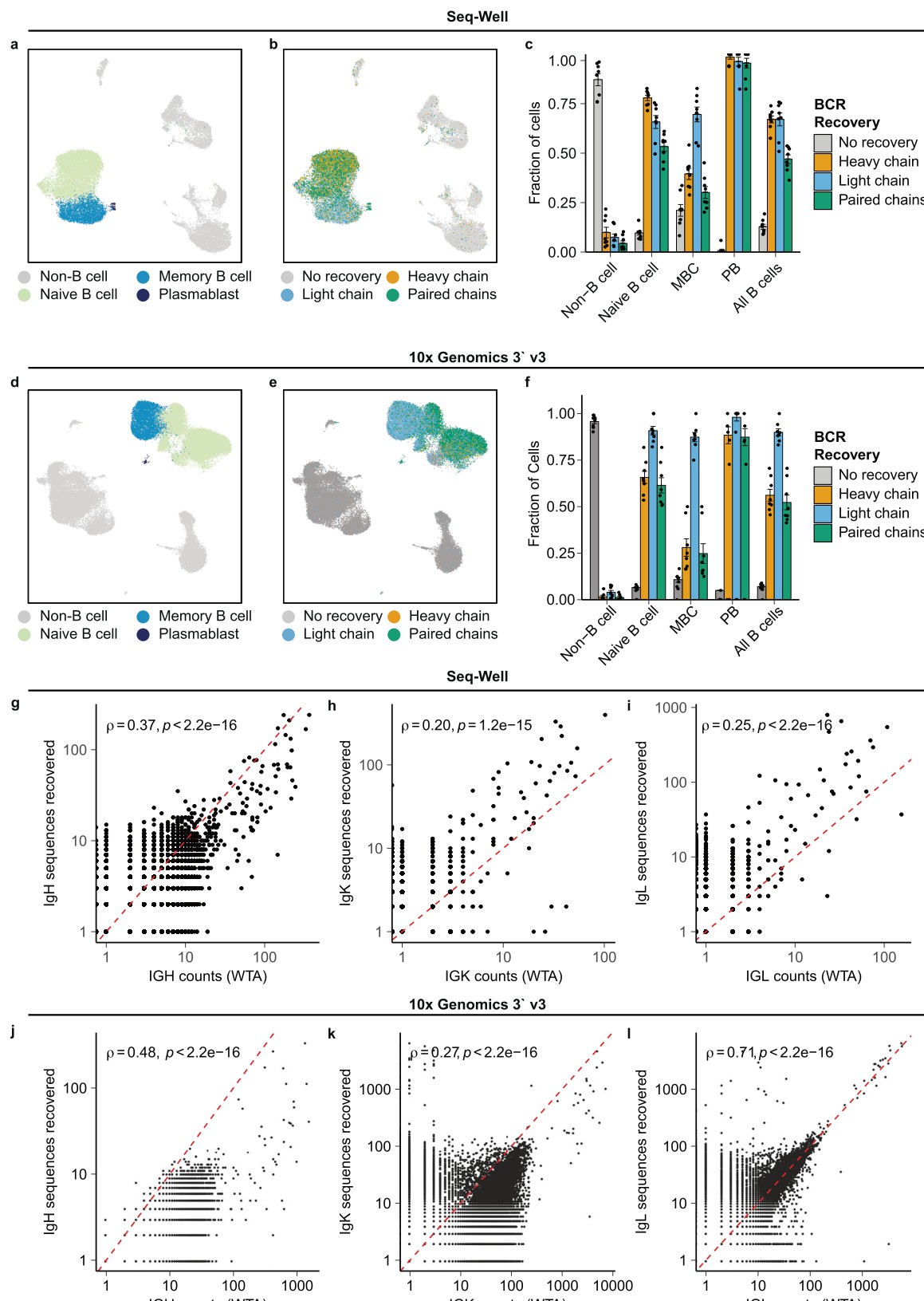

We then recovered BCR sequences from these libraries and achieved similar levels of recovery as in our PBMC experiment (Fig. 4c, d, Supplementary Fig. 5a, b), as well as additional recovery of heavy chain sequences from transfected HEK cells. Many HEK cells from which we did not recover BCR sequences expressed no detectable heavy or light chain transcript in WT libraries, suggesting that overall BCR recovery from transfected HEK cells was limited by poor transfection efficiency or the poor viability of transfected HEK cells relative to non-transfected HEK cells (Supplementary Fig. 5c, d).

To assess whether sequence information was faithfully preserved by our method, we analyzed BCR sequences originating from HEK cells and the U6

**Fig. 2 | Recovery of full-length, paired BCR sequence from human peripheral blood mononuclear cells. a** UMAP of cell phenotypes present in human PBMC prepared using Seq-Well ($n$ = 24,806 cells). **b** UMAP of BCR recovery from single cells prepared using Seq-Well. **c** Fraction of cells with no recovery, recovery of heavy chain, recovery of light chain, and paired recovery using Seq-Well ($n$ = 8 samples). **d** UMAP of cell phenotypes present in human PBMC prepared using 10x Genomics 3' v3 ($n$ = 50,877 cells). **e** UMAP of BCR recovery from single cells prepared using 10x Genomics 3' v3. **f** Fraction of cells with no recovery, recovery of heavy chain, recovery of light chain, and paired recovery using 10x Genomics 3' v3

($n$ = 8 samples). **f–i** Correlation between the number of counts mapping to the IGH/IGK/IGL locus and the number of functional heavy chain or light chain molecules recovered from Seq-Well libraries. Spearman's correlation coefficient and the associated p-value are shown. **j–l** Correlation between the number of counts mapping to the IGH/IGK/IGL locus and the number of functional heavy chain or light chain molecules recovered from 10x Genomics libraries. Spearman's correlation coefficient and the associated $p$ value are shown. For (**c**, **f**), error bars are mean ± standard error of the mean.

BCR. Remarkably, 95.2% of heavy chain sequences and 97.4% of light chain sequences recovered from HEK cells exhibited at most one nucleotide mismatch from the U6 BCR (Fig. 4e). The remaining sequences recovered from HEK cells either exhibited more nucleotide mismatches with the U6 BCR or were unrelated to the U6 BCR. We hypothesize that these unrelated sequences result either from pre-amplification artifacts (i.e., ambient RNA contamination)[48] or the formation of chimeric PCR products during amplification[49]. We also examined the recovery of U6 BCR sequences from all cells and found that greater than 99% of both U6 heavy and light chain sequences were recovered from HEK cells, confirming that our method is reliable at attributing BCR sequences to the correct cells (Fig. 4f).

Lastly, to assess to what extent our method may exhibit a bias in its ability to recover BCRs associated with the use of multiplexed V gene primer sets, we performed bulk VDJ amplicon sequencing of both heavy and light chains of B cells isolated from the same donor, using an approach based on 5'-reversible amplification of cDNA ends (5'-RACE) to avoid introducing any bias that may be associated with amplification with V-region primer sets[50,51]. We found strong correlations between the frequency of heavy and light chain V-region segments in BCRs recovered from single-cell libraries, confirming that the use of V-region primer sets in our approach does not introduce substantial bias into the sequences recovered (Fig. 4g, h).

## Recovery from 3'-barcoded libraries demonstrates comparable recovery to commercially available platforms for recovery from 5'-barcoded libraries

To benchmark the sensitivity of our method against commercially available platforms for the recovery of BCR sequences from 5'-barcoded single-cell libraries, we reanalyzed data generated using 5'-barcoded scRNA-seq (10x Genomics) from three independent studies that performed BCR recovery from samples of human B cells isolated from PBMC or lymphoid tissues (tonsils)[25,28,52]. We first compared the fraction of B cells with recovery of heavy, light, and paired heavy/light chains in these three datasets of 5'-barcoded libraries with the three datasets of 3'-barcoded libraries we generated with Seq-Well or the 10x Genomics 3'-GEX platform (Fig. 5a). We found that the recovery obtained with 5'-barcoded libraries (average across all datasets: 70.0% heavy chain, 82.3% light chain, 68.3% paired chains) was only slightly greater than what we obtained with 3'-barcoded libraries (Seq-Well libraries: 67.8% of heavy chain, 57.0% light chain, and 39.9% paired chains; 10x libraries: 56.1% heavy chain, 89.9% light chain, and 52.2% paired chains).

We then compared the sensitivity of these methods on a molecular basis. We determined the relationship between the number of BCR transcripts enumerated in the WTA product and the probability of BCR recovery from each cell (Fig. 5b). When adjusted for the number of BCR transcripts enumerated in the WTA product, the recovery of heavy chain sequences was similar between 5'-bacoded and 3'-barcoded libraries; our method for 3'-barcoded libraries slightly outperformed the sensitivity of 5'-barcoded libraries for light chain transcripts. This finding suggests that at least some of the trends towards slightly elevated rates of BCR recovery in the 5'-barcoded libraries examined here result from library-intrinsic features, such as the types of B cells in each sample or improved capture of BCR transcripts during library preparation. Overall, these results confirm that our method for recovery of BCR from 3'-barcoded libraries exhibits comparable sensitivity to commercially available solutions that enable recovery from 5'-barcoded libraries.

## Isolation of antigen-specific B cells from rhesus macaques receiving monovalent glycoconjugate pneumococcal vaccines

We next applied our approach to resolve the clonotypic and phenotypic characteristics of antigen-specific B cells elicited by glycoconjugate pneumococcal vaccines. We focused these studies on responses generated against the capsular polysaccharide of *S. pneumoniae* serotype 3 (ST3). We prepared a fluorescently labeled, biotinylated ST3 polysaccharide to label and isolate ST3-reactive B cells. We analyzed PBMC samples from five infant rhesus macaques that were obtained one week after receiving the third dose of a monovalent ST3 glycoconjugate vaccine. We selected these samples from five monkeys, one of which exhibited a strong response to vaccination, as assessed by ST3-specific IgG and opsonophagocytic assay (OPA) titers[53], two that exhibited intermediate responses, and two of which exhibited low responses. As a control, we also analyzed samples obtained from three adult monkeys that had never received pneumococcal vaccines.

From each monkey, we used fluorescence-associated cell sorting (FACS) to isolate ST3-reactive B cells as well as non-ST3-reactive B cells for analysis with scRNA-seq using Seq-Well. We found an increase in the frequency of IgG+ ST3-reactive B cells in vaccinated monkeys relative to the control group, indicating that the majority of IgG+ ST3-reactive B cells are elicited by vaccination (Fig. 6a, b, Supplementary Fig. 6). The small number of ST3-reactive B cells analyzed in unvaccinated monkeys may represent a degree of preexisting humoral immunity acquired through exposure to similar antigens, such as those present in the gut flora[54], though it remains possible that a fraction of these B cells bind to the antigen construct through non-specific means. We also observed modest correlations between the frequency of IgG+ ST3-reactive B cells and ST3-specific IgG and OPA titers, providing support for our strategy to isolate ST3-reactive cells and suggesting that stronger responses to the ST3 antigen are linked to higher frequencies of class-switched, ST3-reactive B cells (Fig. 6c, d).

Across all 8 animals, we recovered single-cell transcriptomes for 6819 cells, including 947 ST3-reactive cells (Fig. 6e). In the resulting transcriptional data, we annotated four clusters of B cells (Fig. 6f, g) including naïve-like B cells, Fos-activated B cells which upregulated transcripts associated with the transcription factor AP-1 (*FOS, FOSB, JUNB*), memory-like B cells (MBC-like cells), and an additional population of memory B cells (B2M^hi) which exhibited downregulation of markers associated with lymph homing (*SELL, CCR7, CXCR5*), elevated levels of BCR inhibitory molecules (*SIGLEC6, SIGLEC10*), and upregulation of molecules associated with tissue homing (*ITGAX, ITGB2, ITGB7*)[25,55]. Compared to B2M^hi B cells, MBC-like cells also upregulated transcripts associated with immunological memory (*BCL2A1*), as well as NF-kB signaling (*NFKBIA, NFKBID, REL, RELB*), while B2M^hi cells upregulated transcripts associated with antigen presentation (*CD74, MAMU-DRA, MAMU-DPA1, MAMU-DOA, MAMU-DRB1*) and with B cell activation (*IGHG1, GPR183, CD1C, MS4A1*) (Fig. 6h, Supplementary Data 1)[56,57].

## Select features of the BCR repertoire are associated with specificity for ST3

To assess the repertoire of ST3-reactive B cells, we next recovered full-length variable region BCR sequences (Supplementary Fig. 7a–c). The degree of BCR recovery achieved was slightly higher among ST3-reactive cells (Supplementary Fig. 7b) but did not differ substantially among transcriptional phenotypes (Supplementary Fig. 7c). Consistent with our cluster annotations, we observed greater levels of SHM frequency among

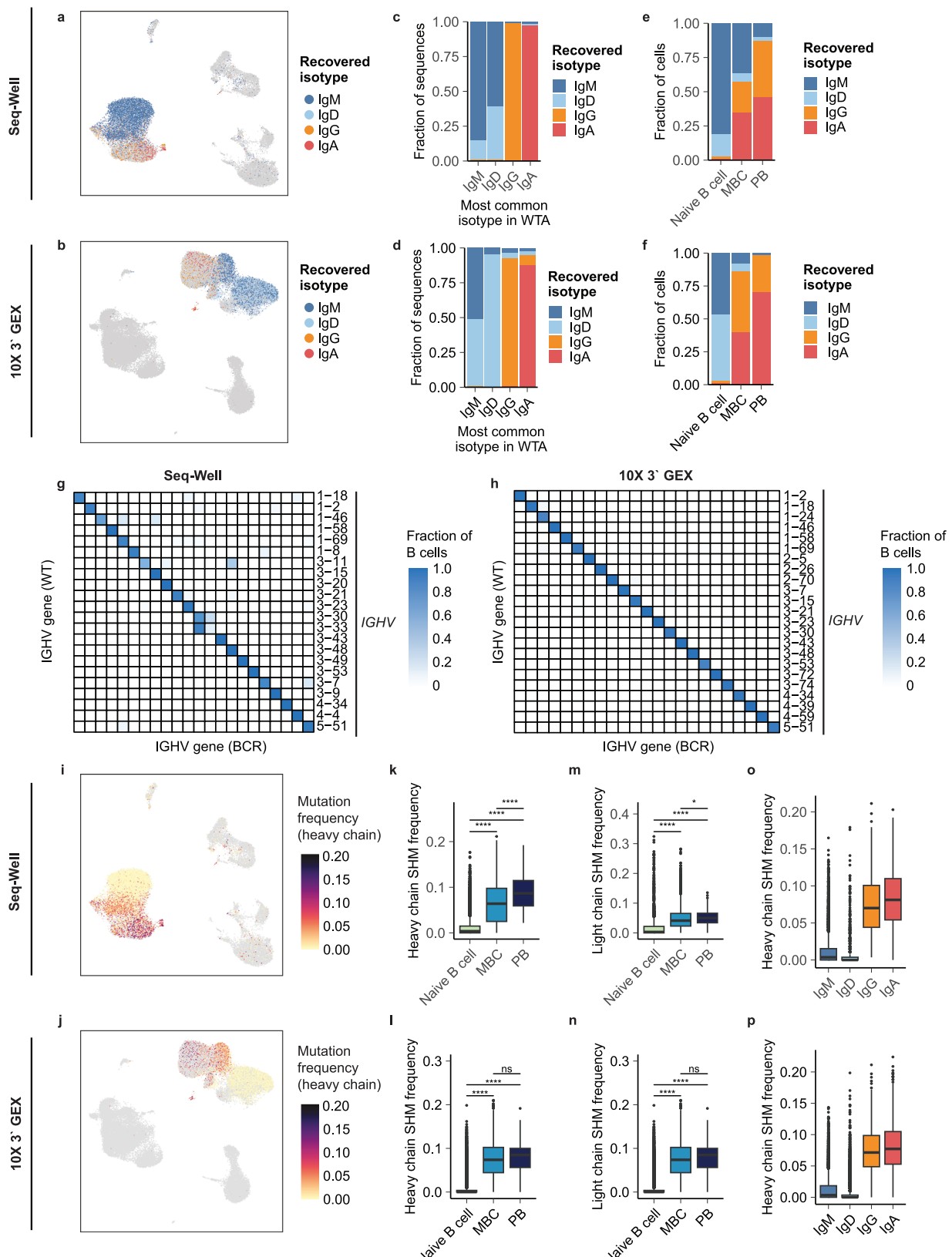

both MBC-like cells and B2M$^{hi}$ cells than naïve cells, as well as increased class switching to IgG among B2M$^{hi}$ cells (Fig. 7a–d, Supplementary Fig. 7d). We observed an increase in the fraction of IgG sequences among ST3-reactive cells in four of the five vaccinated monkeys analyzed, which could indicate enhanced class-switching promoted by vaccination

with the monovalent glycoconjugate vaccine used here (Supplementary Fig. 7e).

Next, we defined clonal lineages using the recovered heavy chain sequences. We found a substantial increase in the frequency and size of expanded clonal lineages recovered from ST3-reactive cells in three of the

**Fig. 3 | Concordance between single-cell whole transcriptome and BCR libraries.** UMAP of isotypes of BCR heavy chain sequences recovered from single-cell libraries prepared by Seq-Well (**a**) and 10x Genomics 3' GEX (**b**), respectively. **c, d** Distribution of recovered BCR isotypes among B cells binned by the most common Ig constant region transcript present in WT. **c** is for Seq-Well-prepared samples, and (**d**) for 10x Genomics 3' GEX. All cells shown have at minimum 3 counts of transcript detected in the WTA product. sotypes of BCR heavy chain sequences recovered from single-cell libraries, grouped by phenotypes assigned in single-cell gene expression data from Seq-Well (**e**) and 10x Genomics 3' GEX (**f**), respectively. All genes shown have at least two transcripts recovered in both WT and BCR libraries. **g, h** Heat map comparing most common heavy chain V gene segments in WT libraries and V-genes of recovered BCR sequences. Heavy chain somatic mutation frequency overlaid onto UMAP of cells prepared by Seq-Well (**i**) and 10x Genomics 3' GEX (**j**), respectively. **k, l** Somatic mutation frequency of heavy chain BCR sequences grouped by B cell phenotypes. **m, n** Somatic mutation frequency of light chain BCR sequences grouped by B cell phenotypes. **k, m** are for Seq-Well-prepared samples, and (**n**) for 10x Genomics 3' GEX. **o, p** Somatic mutation frequency of BCR sequences grouped by B cell isotypes. *P* values are calculated using a two-sided Wilcoxon rank-sum test and are adjusted using Bonferroni correction. ns $p > 0.05$, $*p <= 0.05$, $**p <= 0.01$, $***p <= 0.001$, $****p <= 0.0001$.

five vaccinated monkeys, consistent with these cells having recently undergone expansion in response to antigen encounter (Fig. 7e). Expanded clonotypes in other cells were likely not detected in the other two vaccinated monkeys due to the low number of ST3-reactive cells from these monkeys for which BCR information was recovered (Supplementary Fig. 7f). All expanded clonal lineages were detected exclusively in either the ST3+ or ST3- fractions, supporting that reactivity to the ST3 antigen is shared by cells in a clonotype and suggesting that our cell isolation strategy successfully isolated ST3-specific B cells from non-ST3-specific B cells (Supplementary Fig. 7g). Interestingly, two expanded ST3+ clonotypes were detected in a single monkey from the unvaccinated control group (Supplementary Table 2). These clonotypes expressed IgM isotypes but had BCRs that were not in germline configuration, suggesting that they had undergone mutation in response to prior exposure to a similar antigen.

We next aimed to define BCR motifs associated with specificity for the ST3 antigen. First, we compared the frequency of V gene segments utilized by heavy and light chain sequences between ST3-reactive cells from the vaccine group and non-ST3-reactive cells. We determined that five IGHV genes and seven IGKV or IGLV genes were statistically enriched (Bonferroni-adjusted $p < 0.001$) in frequency among ST3+ cells, suggesting that these gene segments encode motifs that promote specificity to the ST3 antigen (Fig. 7f, g). Together, this set of V genes comprised 62.3% of all ST3-reactive heavy chain sequences and 75.3% of ST3-reactive light chain sequences recovered.

To further define motifs associated with ST3 reactivity, we grouped ST3-reactive sequences by these IGHV genes and analyzed the lengths of their CDRH3 junctions and the usage of IGKV/IGLV genes by their paired light chains (Fig. 7h, i). We found a clear preference for CDRH3 length and IGKV/IGLV genes by ST3-reactive clonotypes utilizing each of these IGHV genes, suggesting that these combinations of BCR features are associated with reactivity towards ST3. For the three IGHV genes for which we recovered the greatest number of unique clonotypes, we further analyzed CDRH3 junctions and identified further sequence characteristics associated with reactivity to ST3, including: "GSY" at codons 4-6 of the CDRH3 junction of ST3-reactive clonotypes using *IGHV4-173*, "Y" at codon 4 of clonotypes using *IGHV4-80*, and "L(V/I)G" at codons 9-11 of clonotypes using *IGHV4-99* (Fig. 7j). Altogether, they suggest an enrichment and selection of BCR features that are associated with reactivity towards ST3.

**Rhesus macaques exhibit a convergent repertoire in response to vaccination**

We next assessed to what extent these sequence motifs were utilized by each of the vaccinated monkeys. Remarkably, all five IGHV genes identified here as associated with ST3-reactivity were utilized by clonotypes present in two or more monkeys in the vaccine group, and all five vaccinated monkeys possessed clonotypes with at least one of these IGHV genes, indicating a convergence for a defined set of ST3-reactive sequences in monkeys receiving this pneumococcal vaccine formulation (Fig. 7k). Interestingly, we also identified clonotypes that utilized four of these five IGHV genes among ST3+ cells from two of the three unvaccinated monkeys. We further examined these potentially ST3-reactive sequences from unvaccinated monkeys and found that, while clonotypes utilizing *IGHV4-127*, *IGHV4-80*, and *IGHV4-99* from these unvaccinated monkeys exhibited preferences for

distinct CDRH3 junction lengths and IGKV/IGLV gene pairings than corresponding ST3-reactive sequences from vaccinated monkeys, a subset of *IGHV4-173* ST3-reactive sequences from unvaccinated monkeys exhibited a similar preference for CDRH3 junctions of 11 or 12 amino acids as well as pairing with *IGKV2-65* or *IGLV2-23*, demonstrating a level of similarity with ST3-reactive clonotypes recovered from vaccinated monkeys (Supplementary Fig. 8a, b). Consistent with this observation, a recent study of patients receiving the 23-valent Pneumovax polysaccharide pneumococcal vaccine demonstrated that the vaccine resulted in the expansion of pre-existing capsular polysaccharide-specific plasma cells that shared reactivity with antigens present in the gut flora[54].

Lastly, to verify that the BCR motifs identified here were specific for ST3, we selected a total of 24 clonotypes to express as Rhesus-murine IgG1 chimeric antibodies. Of these, 23 expressed and were used to evaluate specificity against 93 different pneumococcal serotypes (Fig. 8a, b, Supplementary Data 2). Remarkably, 20 of these 23 clonotypes exhibited strong, specific binding to ST3, and lacked substantial levels of binding to other pneumococcal serotypes. Of the three remaining clonotypes, two exhibited a weak level of polyreactivity to a majority of pneumococcal serotypes evaluated, and one exhibited minimal to no binding to any of the polysaccharides evaluated. We hypothesize that these antibodies may lack reactivity to ST3 in this assay due to structural variations when expression in the murine backbone or due to propagated errors in the original BCR sequence.

## Discussion

Here, we developed a method (B3E-seq) to recover full-length, paired BCR sequences from 3'-barcoded gene expression scRNA-seq libraries. We demonstrated that B3E-seq accurately matches the single-cell transcriptomes and uses inexpensive, widely available reagents and platforms. Importantly, our approach can be used for both new and archived 3'-barcoded libraries.

Using this method, we analyzed B cell responses elicited against the ST3 antigen by glycoconjugate vaccines. We revealed clonotypic features associated with specificity against the ST3 antigen, and we showed that these features emerge in clonotypes recovered from distinct vaccinated monkeys, demonstrating a convergence towards a common set of antigen-specific motifs. We hypothesize that the convergence of these BCR paratopes reflects the recognition of common, immunodominant epitopes on the ST3 antigen, similar to what has been observed in other contexts[18,58–62]. Interestingly, the simple structure of the ST3 antigen, compared to the polysaccharide capsules of other pneumococcal serotypes, may influence the degree of repertoire convergence by exposing a limited number of epitopes[63,64]. We expect that further studies assessing the importance of these clonotypic and transcriptional features in providing long-lasting immune protection against ST3 or comparing these results to those generated with vaccines against other pneumococcal polysaccharides will be informative and interesting.

Our method has at least two practical limitations. First, it does not recover paired, full-length BCR for every B cell in a pooled scRNA-seq product. This likely derives from limitations in the efficiency of RNA capture inherent to the scRNA-seq platforms. It is well-established that scRNA-seq libraries contain only a fraction of the total mRNA content of a single cell[35,65,66], and here we have consistently observed correlations between levels

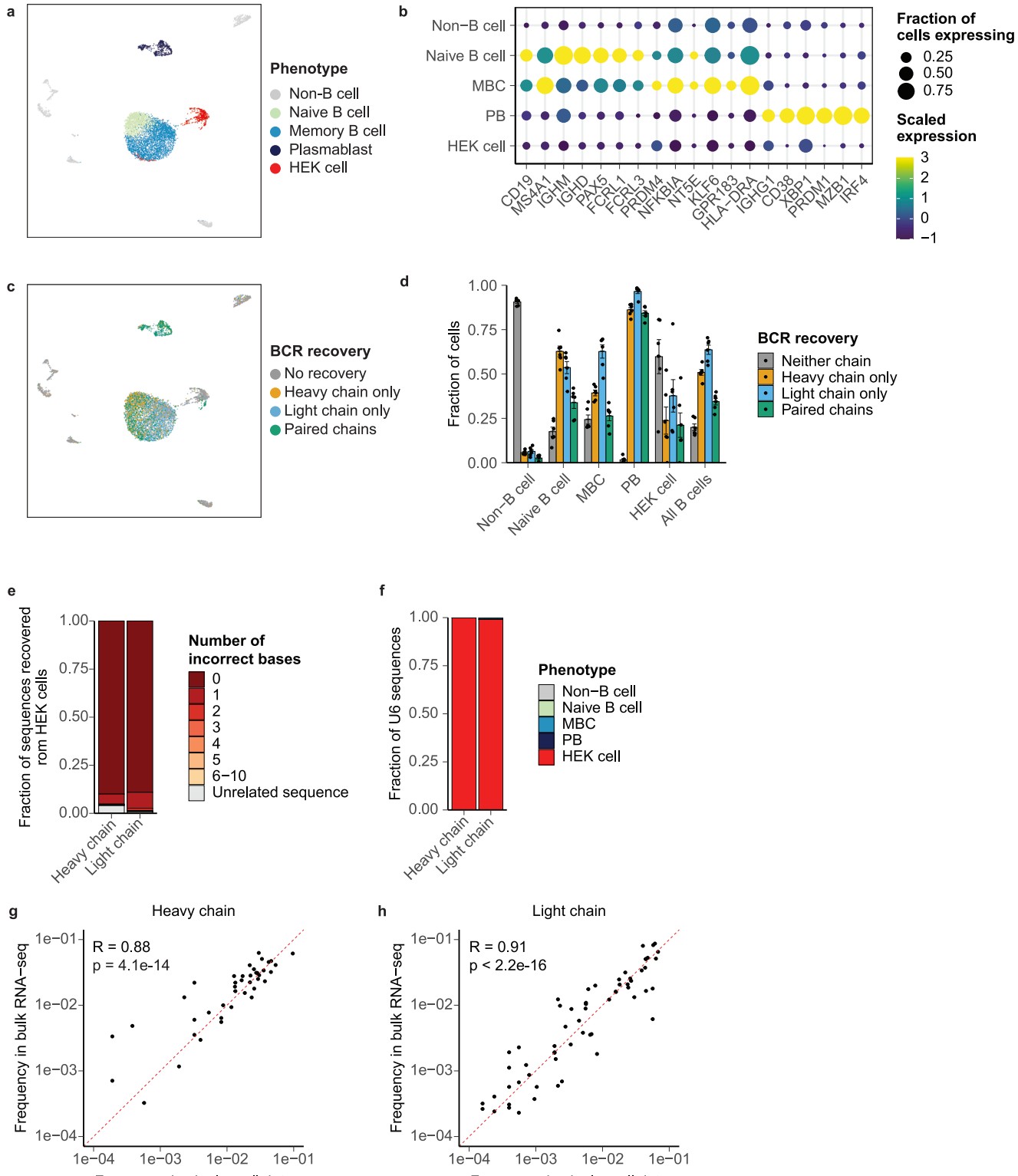

**Fig. 4 | Accuracy and sensitivity of the approach. a** UMAP of transcriptional phenotypes present in B cell-enriched PBMC with spike-in of U6-transfected HEK cell line. **b** Dot plot showing scaled expression and the fraction of cells expressing B cell marker genes. **c** UMAP of BCR recovery. **d** Fraction of BCR recovered from each transcriptional phenotype. **e** Relationship of heavy and light chain sequences recovered from HEK cells to U6 BCR. **f** Cell phenotypes with recovery of U6 BCR. Correlation between *IGHV* (**g**) and *IGKV/IGLV* (**h**) frequency in recovered BCR from single cells and BCR from 5'-RACE bulk sequencing. Pearson's correlation coefficient and the associated *p* value are shown.

of BCR transcript expression and the probability of BCR recovery from the same cell (Fig. 2g). Thus, future advances in the molecular biology used to prepare scRNA-seq libraries could enhance the sensitivity of our approach. Additionally, our method requires species-specific elements, including the

multiplexed V-region primer sets used for primer extension. The design of these primers requires the accurate annotation of V genes for a given organism, and an understanding of the frequencies of somatic mutations that are likely to occur that may affect the binding of these primers. While

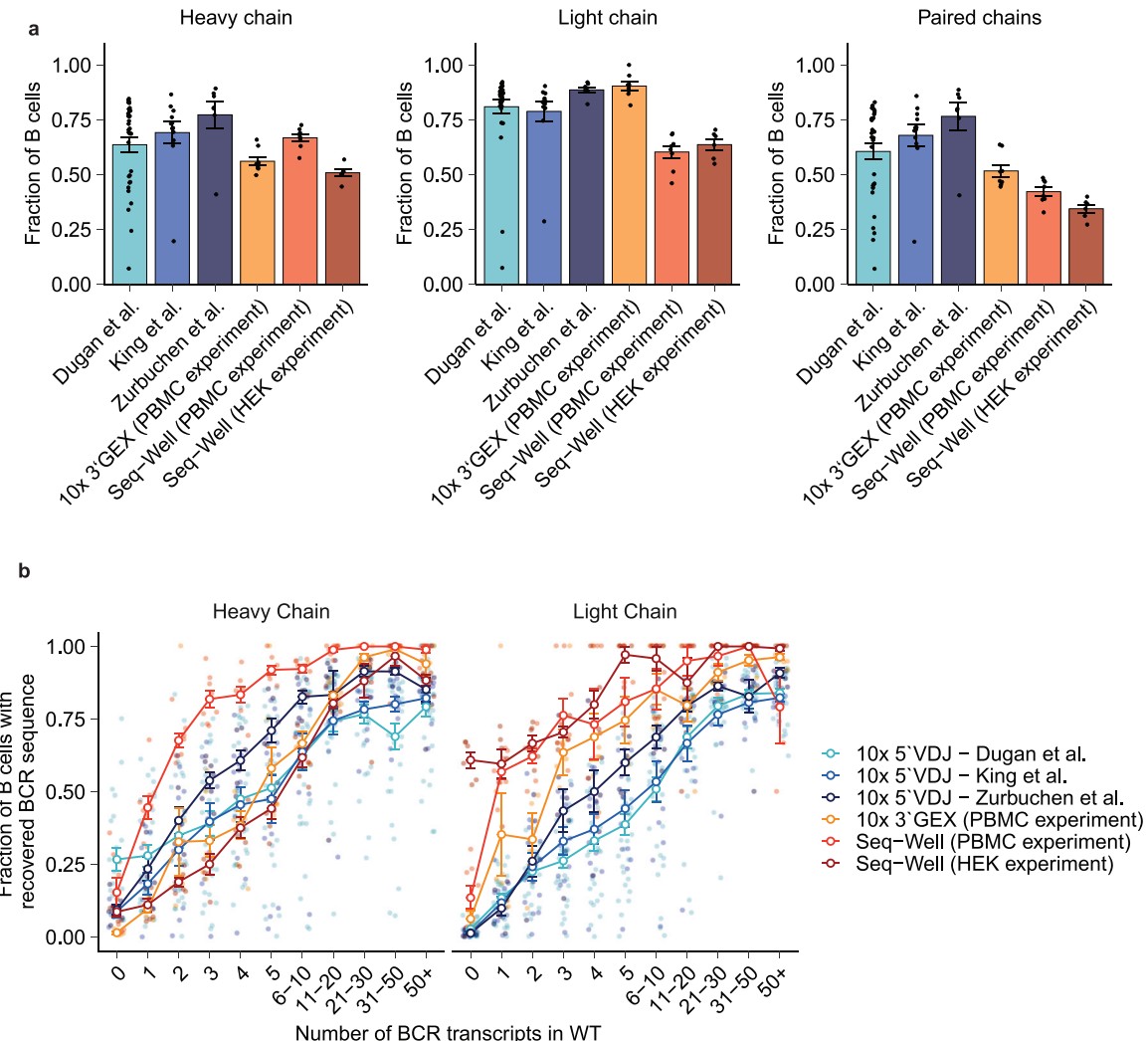

**Fig. 5 | Comparison of BCR recovery to 5'-barcoded single-cell platforms. a** Rates of recovery of heavy chain, light chain, and paired chain BCR from either 5'- or 3'-barcoded datasets. Each point represents a single experimental replicate. **b** Relationship of the average probability of BCR recovery and the number of heavy chain or light chain transcripts enumerated in whole transcriptome sequencing. Each jitter point represents a reaction lane of 10x Chromium or Seq-Well array. The error bars are mean ± standard error of the mean.

primer sets for many commonly studied model organisms are available[67–71], this requirement may prove a barrier to studying less well-characterized organisms. Nevertheless, the amount of publicly available immunoglobulin repertoire data is increasing, and tools for discovering germline immunoglobulin V genes from repertoire sequencing data have become increasingly more sophisticated[72–74], enabling the definition of novel or personalized germline allele databases with greater accuracy and ease. As a result, the characterization of V genes in many species continues to improve, and we expect the availability of suitable primer sets to expand in the future[69,73,75].

In summary, our method recovers BCR sequences that are accurate and cgioncordant with the transcriptional profiles of the same cells, and the sensitivity of the method is limited predominantly by the efficiency of mRNA capture and level of BCR expression. Our method is compatible with 3'-barcoded scRNA-seq library produced with both Seq-Well and 10x Genomics, and we anticipate that it can be adapted to a broad range of single-cell library platforms, including emerging multi-modal approaches that aim to quality surface epitope expression, chromatin accessibility, DNA methylation, and/or spatial location in combination with single-cell transcriptomes[76–81]. By applying the method to antigen-specific B cells

elicited by pneumococcal vaccination, we demonstrate a convergence of BCR features among distinct monkeys. We anticipate that B3E-seq will be especially useful in analyzing the relationships between phenotypic and clonotypic features of antigen-specific B cell populations in vaccination or disease.

## Methods

### Ethics statement

Immunizations for the non-human primate study were performed at the University of Louisiana at Lafayette-New Iberia Research Center (NIRC), which is accredited by the Association for Assessment and Accreditation of Laboratory Animal Care (AAALAC, Animal Assurance #: 000452). The work was in accordance with the USDA Animal Welfare Act and Regulations and the NIH Guidelines for Research Involving Recombinant DNA Molecules, and Biosafety in Microbiological and Biomedical Laboratories. All procedures performed on these animals were under regulations and established guidelines and were reviewed and approved by an Institutional Animal Care and Use Committee or through an ethical review process. We have complied with all relevant ethical regulations for animal use. All ethical regulations relevant to human research participants were followed.

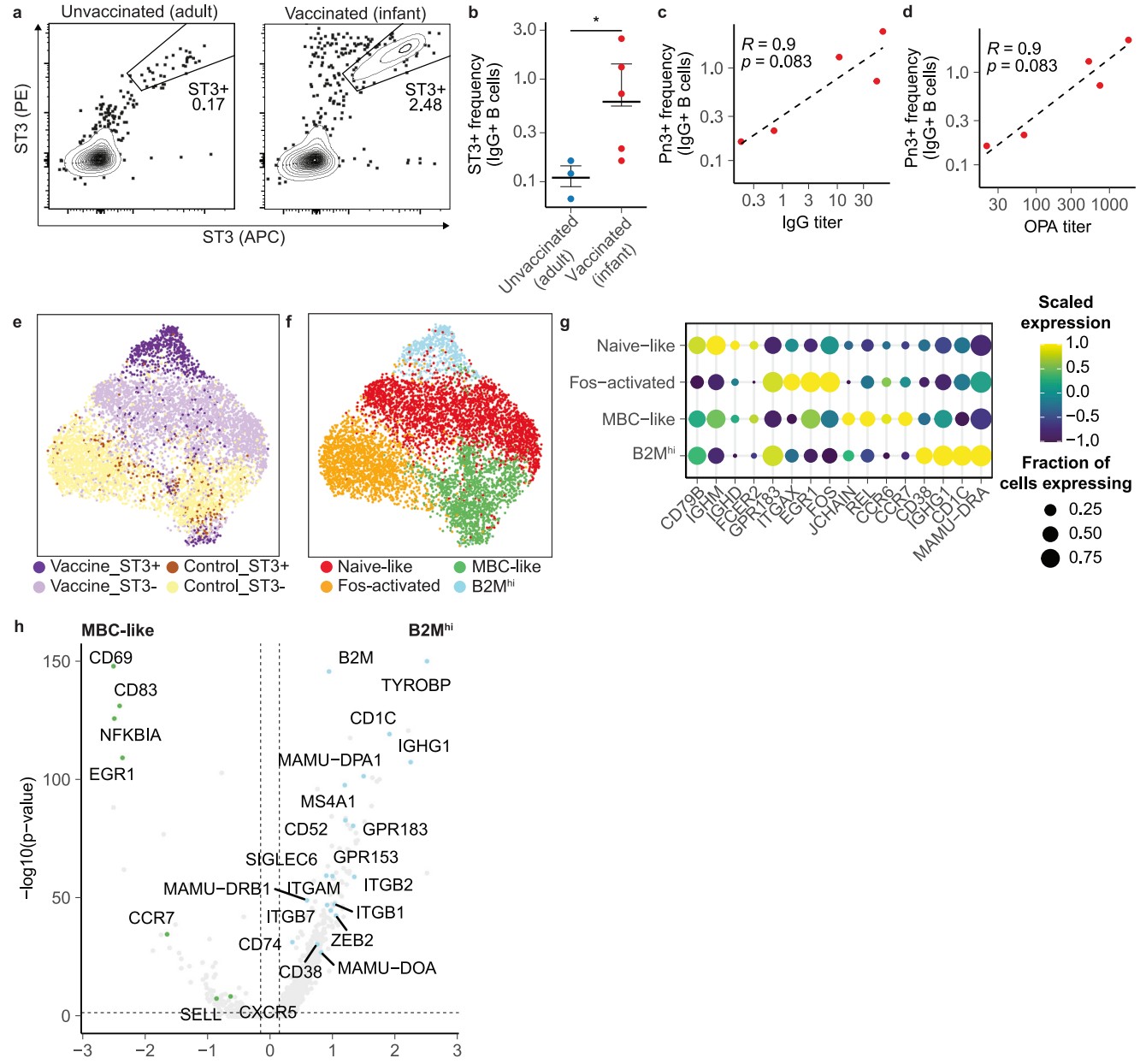

**Fig. 6 | Transcriptional features of ST3-reactive and non-ST3-reactive B cells elicited by vaccination. a** Representative staining of ST3-reactive B cells from unvaccinated and vaccinated samples, gated on IgG+ B cells (Live CD3⁻ CD19⁺ CD20⁺ IgG⁺). **b** Frequency of ST3-reactive IgG+ B cells in vaccinated and unvaccinated monkeys. The *P* value is calculated using a one-sided Wilcoxon rank-sum test. *$p \leq 0.05$. Correlation between ST3-reactive IgG+ B cells and ST3-specific IgG (**c**) and OPA (**d**) titers. Spearman's correlation and the associated *p* value are shown.

UMAP of single-cell transcriptomes colored by sort fraction (**e**) and cell phenotype (**f**). **g** Dot plot showing scaled expression and percent of cells expressing B cell marker genes in each transcriptional phenotype. **h** Volcano plot of differentially expressed transcripts between B2M^hi and MBC-like cells. *P* values are calculated using a two-sided Wilcoxon rank-sum test and are adjusted using Bonferroni correction.

## Animals and immunization

Infant rhesus macaques were housed with their moms for the duration of the study at the New Iberia Research Center (NIRC), University of Louisiana. The infant rhesus macaque experimental protocol was approved by IACUC at both Pfizer Inc. and NIRC. Age (3–6 months old at the start of the study) and sex-matched infant rhesus macaques were randomly divided into groups for vaccination. Infants were vaccinated under sedation with 2.2 µg of pneumococcal serotype 3 conjugate vaccine (ST3) formulated with 125 µg of AlPO₄ at 0.25 ml volume of final formulation intramuscularly (IM) into a single limb on day 0. Pre-bleeds to assess baseline ST3-specific sera titers were collected 1 week (wk = −1) before primary vaccination (D0).

Post-vaccination blood for sera was collected at 4 weeks post-dose 1 (PD1). Control serum and PBMC samples were collected from adult Rhesus Macaques at NIRC from naïve animals that had not been previously vaccinated with pneumococcal vaccines.

## Determination of ST3-specific IgG and OPA titers

*Streptococcus pneumoniae* direct Luminex immunoassays (dLIAs) were used to quantify IgG antibody concentrations in serum for ST3 polysaccharide. This was done using Luminex Magplex technology (Luminex Corporation, Austin, TX). Briefly, serum samples, reference standard serum, and QCS prepped by Hamilton Robotic units (Hamilton Company,

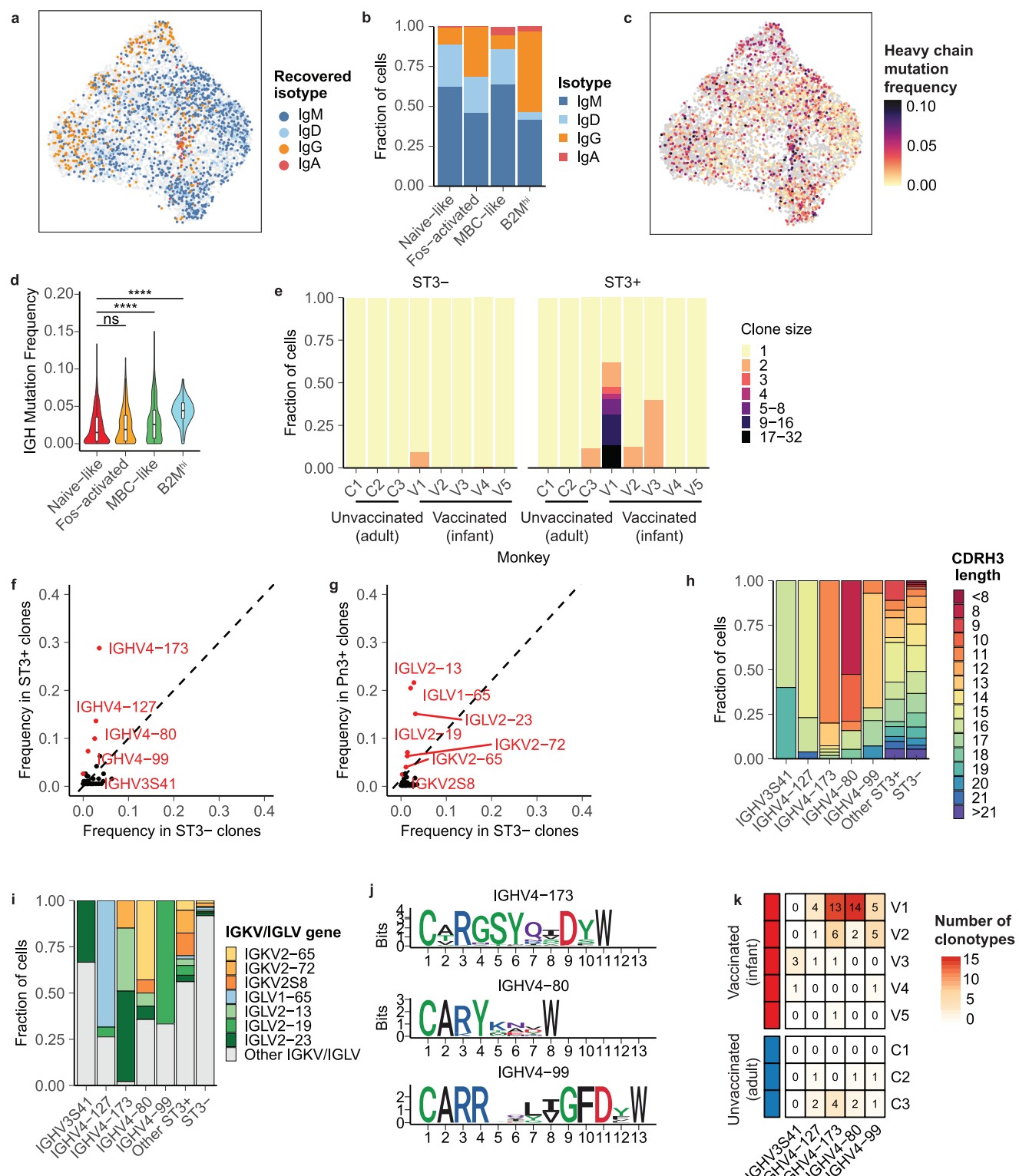

**Fig. 7 | Clonotypic features of ST3-reactive B cells. a** UMAP colored by isotype of recovered BCR. **b** Isotypes recovered from B cell phenotypes. **c** UMAP colored by frequency of somatic mutation. **d** Frequency of somatic mutation in heavy chain sequences recovered from each phenotype. P-values are calculated with a two-sided Wilcoxon rank-sum test and are adjusted with Bonferroni correction. **e** Clonal sizes of ST3-reactive and ST3- B cells. Frequency of IGHV (**f**) and IGKV/IGLV (**g**) genes among ST3+ cells from vaccinated monkeys and all ST3- cells. Genes highlighted in red are statistically significantly enriched in the ST3+ fraction ($p < 0.001$, $p$ values calculated using a two-sided chi-squared test and are adjusted using Bonferroni correction). **h** CDRH3 junction lengths of ST3-reactive cells using *IGHV* genes statistically associated with ST3-reactive cells. **i** IGKV/IGLV gene pairings lengths of ST3-reactive cells using *IGHV* genes statistically associated with ST3-reactive cells. **j** Logo plots of CDRH3 junctions of ST3-reactive clonotypes using *IGHV4-173*, *IGHV4-80*, and *IGHV4-99*. **k** Number of clonotypes with each V-gene recovered from each vaccinated and unvaccinated monkey.

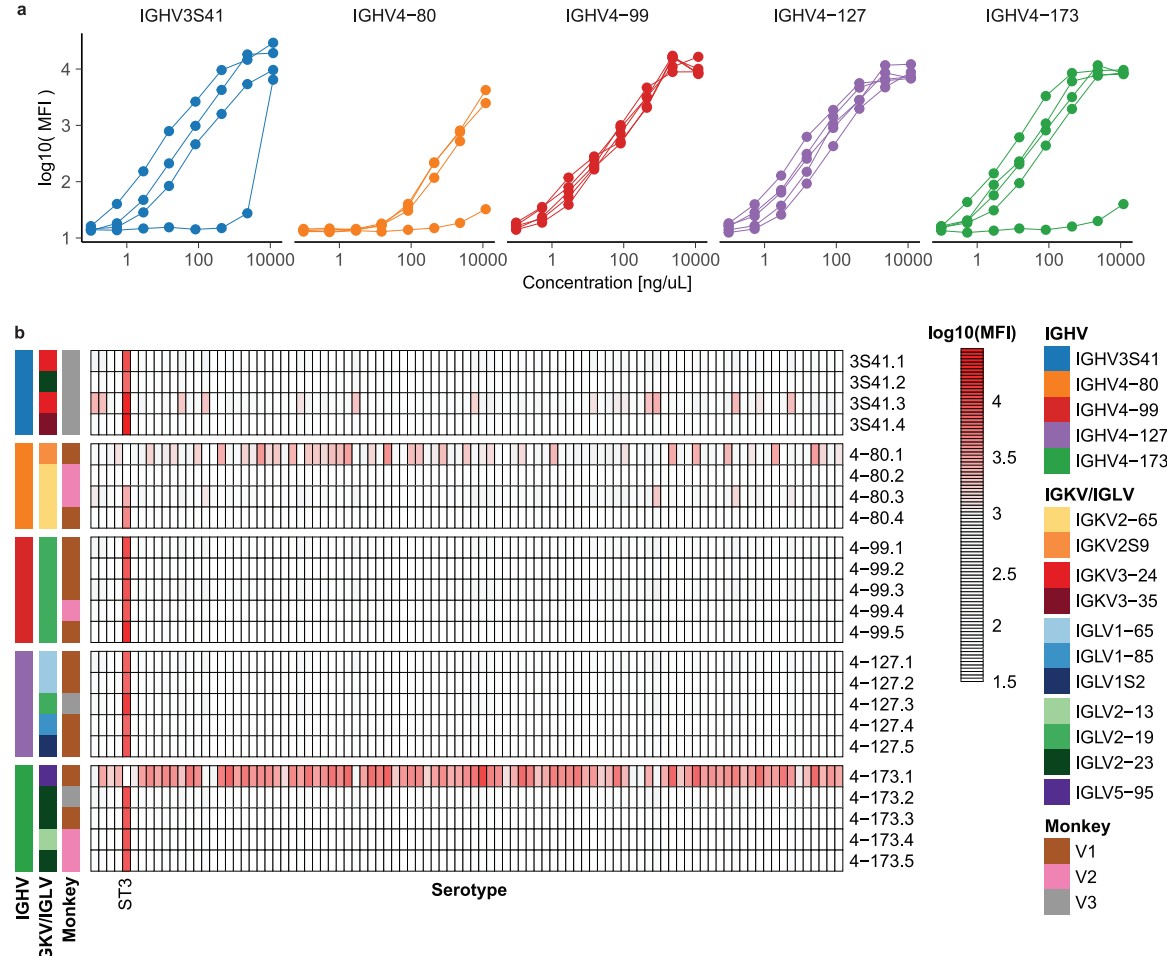

**Fig. 8 | Convergent BCR sequences exhibit specific binding to ST3. a** Plots of recombinant Rhesus-murine chimeric antibodies binding to ST3 as a function of concentration and IGHV gene. Each line represents an antibody selected from a single clone. **b** Heatmap of antibody binding to 93 different pneumococcal serotypes as a function of IGHV gene family and monkey. The values are shown for MFI measured with 2.3 μg/L of antibody by Luminex. The full list of serotypes evaluated is provided in the Supplementary Data 2.

Reno, NV). PnPS-coupled magnetic microspheres were added to each well of the assay plates and incubated. After 90 min non-bound components were washed off and the secondary antibody (R-Phycoerythrin-conjugated goat anti-human IgG) was added to each well. After 90 min the assay plate was washed again and 100 μl of LXA-20 was added to each well. After 2 h of incubation, the plates were then loaded into Bio-plex 200, and median fluorescent intensities (MFI) were read. These MFIs were used to calculate antigen-specific IgG concentrations (μg/ml) against established IgG concentrations of the reference standard by statistical analysis software.

Microcolony OPA is based on antibody-mediated complement- and phagocyte-dependent OPA assays. Here, OPA assays were developed for S. pneumoniae serotype 3. Briefly, heat-inactivated sera were serially diluted. Target bacteria were added to assay plates and incubated. Baby rabbit complement and differentiated HL-60 cells were added to each well at an approximate effector-to-target ratio. After incubations, resulting colonies were stained with Coomassie Brilliant Blue stain. Colonies were imaged and enumerated on a CTL ImmunoSpot Analyzer®. The interpolated OPA antibody titer is the reciprocal of the dilution that yields a 50% reduction in the number of bacterial colonies when compared to control wells that did not contain serum.

**Preparation of biotinylated ST3 antigen**
Pneumococcal serotype 3 polysaccharide was mixed with imidazole (3x, w/w) and pH was adjusted to 3.5 with 1M-HCl, then frozen and lyophilized. After lyophilization, the lyophilized polysaccharide was reconstituted with

anhydrous DMSO to make 4 mg/mL saccharide concentration. The reaction mixture was then warmed to 35 °C, and CDI (0.1 MEq) was added subsequently. The reaction mixture was stirred at 35 °C for 3 h. After the reaction mixture was cooled to 23 °C, WFI (2% v/v) was added to quench free CDI and stirred further for 30 min at 23 °C. Biotin hydrazide (1 MEq) was added. The reaction mixture was stirred at 23 °C. After 20 h reaction, the reaction mixture was diluted to chilled (at 5 °C) PBS buffer (5X, v/v). The diluted reaction mixture was then purified by UF/DF using 10 K MWCO PES membrane (Millipore Pellicon 2 Mini) against PBS buffer (30X, v/v) and filtered through 0.22 μm and analyzed.

**PBMC isolation**
Venous blood samples were collected in heparinized tubes. Each sample was diluted 1:1 with 0.9% NaCl and applied to Accuspin tubes (Sigma-Aldrich, Burlington, MA) filled with a Ficoll–Paque solution (ThermoFisher Scientific, Waltham, MA) before centrifugation. PBMCs were recovered from the interface, washed once in 1X HBSS Cellgro (ThermoFisher), resuspended in Recovery Cell Culture Freezing Medium (Gibco) at a concentration of $5 \times 10^6$ cells/ml, and frozen in liquid nitrogen for future use. All ethical regulations relevant to human research participants were followed.

**ST3-PE and ST3-APC coupling**
Biotinylated pneumococcal serotype 3 polysaccharide with a 5% biotin load was coupled with high-concentration streptavidin-phycoerythrin (SA-PE) and streptavidin-allophycocyanin (SA-APC, both from BioLegend, San

Diego, CA) in separate reactions. Biotinylated ST3 and the fluorophore-conjugated streptavidin reagents were prepared with PBS at 4X working solutions of 20 µg/mL and 200 µg/mL, respectively. Equal volumes of ST3 and SA-PE; and, separately, equal volumes of ST3 and SA-APC working solutions were combined, thoroughly mixed, and incubated at 4 ℃ protected from light for a minimum incubation period of 60 min with thorough mixing at 30 min and directly prior to use. Coupled ST3-PE and ST3-APC solutions were maintained at 4 ℃, and protected from light until use.

## Magnetic isolation of B cells from human PBMC

Whole human blood was obtained from Research Blood Components, LLC (Watertown, MA), and PBMCs were isolated using density gradient centrifugation and cryopreserved. Upon use, PBMCs were thawed into Aim-V medium (ThermoFisher). B cells were isolated using an EasySep Human Pan-B Cell Enrichment kit (Stemcell Technologies, Vancouver, Canada).

## Plasmid generation

Monoclonal antibody U6 heavy and light chain vectors were created from an affinity-selected Ara h 2-reactive B cell from a patient undergoing peanut oral immunotherapy (NCT01324401)[17]. Briefly, peripheral blood cells were isolated by Ficoll gradient (GE Healthcare) and stained with an Ara h 2-Alexa Fluor 488 multimer and fluorescent antibodies, CD3–APC (eBioscience #17-0032-82), CD14-APC (eBioscience #17-0149-42), CD16-APC (eBioscience #17-0168-42), CD19–APC-Cy7 (BD Biosciences #561743), CD27–PE (BD Pharmingen #560985), CD38–BV421 (BD Biosciences #562444), and IgM–PE-Cy5 (BD Pharmingen #551079), for single-cell sorting on a FACS Aria II sorter (BD Biosciences) into a 96 well plate. Heavy and light chains underwent nested amplification using previously published primer sets[17] followed by Sanger sequencing (Genewiz). Productive sequences were cloned into a heavy chain IgG1 vector and a light chain kappa vector using restriction enzymes AgeI, BsiWI, SalI, and XhoI (New England Biolabs) for transformation with competent Escherichia coli NEB5a bacteria (New England Biolabs) and colony selection. Plasmid DNA from overnight liquid cultures (LB with 100 mg/mL ampicillin) was isolated with QIAprep Spin Miniprep Kit (Qiagen).

## HEK cell culture and transfection

Plasmid DNA (25 ng) was transfected into HEK293 cells (ATCC, CRL-3216) by using GenJet In Vitro DNA Transfection Reagent (SignaGen, Rockville, Md) and cultured overnight at 37 ℃ with 5% $CO_2$ in serum-free HL-1 media (Lonza, Walkersville, Md) supplemented with Pen-Strep and 8 mmol/L Glutamax (Gibco, Carlsbad, Calif).

## Flow cytometry and cell sorting

Cryopreserved PBMC were thawed in AIM-V media and washed twice with PBS. Tetramer reagent was prepared by combining equal volumes of streptavidin-PE or streptavidin-APC solutions (both from BD) at 200 µg/mL with polysaccharide-biotin solution at 20 µg/uL. Cells were then stained with eBioscience Fixable Viability Dye eFlour 780 (Thermo-Fisher #65-0865-14) and Fc block (BD) on ice for 30 min. Cells were then washed twice with PBS and then stained with 25 µL of each ST3-PE tetramer and ST3-APC tetramer on ice for 30 min. Cells were then washed three times in FBS staining buffer (BD) and were then stained with BV605-conjugated anti-IgM (BioLegend #314523), BV785-conjugated anti-CD20 (BioLegend #302356), DyLight 405-conjugated anti-CD19 (CB19) (Fisher Scientific #NBP225196V), V500-conjugated CD3 (SP34-2, BD Biosciences #560770), PE-Cy5-conjugated anti-IgG (G18-145, BD Biosciences #551497), and Total-seq A anti-human hashtag antibody (Biolegend) on ice for 30 min. Live+CD3-CD19 + CD20 + ST3+ and Live+CD3-CD19 + CD20 + ST3- cells were sorted on a BD Aria 3 sorter, and data was analyzed using Flowjo v10.

## Single-cell RNA-sequencing

Single-cell suspensions were processed for scRNA-seq using the Seq-Well platform with second-strand chemistry[32,33] or the 3′ GEX kit from 10x

Genomics. Libraries were barcoded and amplified using the Nextera XT kit (Illumina, San Diego, CA) and were sequenced on a Novaseq 6000 (Illumina).

## Analysis of single-cell whole-transcriptome data

Raw read processing of scRNA-seq reads was performed using the Drop-Seq processing pipeline[35]. Briefly, reads were aligned to the hg38 reference genome or Mmul_10 reference genome and collapsed by cell barcode and UMI. Whole transcriptome data was further analyzed in Seurat[82]. First, cells with less than 300 unique genes detected and genes detected in fewer than 5 cells were filtered out. Variable features were then determined using the FindVariableFeatures function, and the ScaleData function was then used to regress out the number of RNA features in each cell. The number of principal components used for visualization was determined by examination of the elbow plot, and two-dimensional embeddings were generated using uniform manifold approximation and projection. Clusters were determined using Louvain clustering, as implemented in the FindClusters function in Seurat. For analysis of ST3-reactive B cell data, clusters of cells that exhibited high expression of mitochondrial genes or markers associated with other cell phenotypes (i.e., monocytes) were moved and the data was reprocessed. Differential gene expression analysis was performed for each cluster and between indicated cell populations using the FindMarkers function.

## Enrichment of BCR transcripts

Enrichment of BCR transcripts from whole transcriptome amplification products was performed using xGen Lockdown reagents (IDT). The detailed protocol is included in the Supplementary Protocol. In brief, biotinylated probes for IGHM, IGHD, IGHG, IGHD, IGHE, IGKC, and IGLC were synthesized by IDT and were used at a concentration of 1.5 µM (each oligo) (Supplementary Data 3). For samples generated with Seq-Well, 3.5 µL of WTA product was combined with 8.5 µL 2x hybridization buffer, 2.7 µL hybridization buffer enhancer, 0.5 µL human cot-1 DNA (IDT), and 0.8 µL Seq-Well WTA primer (50 µM). For samples generated with 10x Genomics 3'GEX, 3.5 µL of WTA product was combined with 8.5 µL 2x hybridization buffer, 2.7 µL hybridization buffer enhancer, 0.5 µL human cot-1 DNA (IDT), 0.4 µL 10x forward primer (100 µM), and 0.4 µL 10x reverse primer (100 µM). The mixers were combined and incubated at 95 ℃ for 10 min. Then, 1 µL of hybridization probe mix was added. The mixture was incubated at 65 ℃ for 1 h and was then processed according to the remainder of the xGen lockdown protocol. 50 µL of streptavidin M270 Dynabeads (ThermoFisher) were used for each sample. At the end of this protocol, products were eluted in 20 µL.

The enriched product was then amplified with PCR. Five PCR reactions for each enriched sample were performed with the following composition per reaction: for samples generated with Seq-Well, 12.5 µL of 2x Kapa Hifi Hotstart Readymix (Roche), 8.5 µL water, 2.0 µL Seq-Well WTA primer (10 µM), and 2.0 µL of BCR-enriched product; for samples generated with 10x Genomics 3'GEX, 12.5 µL of 2x Kapa Hifi Hotstart Readymix (Roche), 8.5 µL water, 1.0 µL 10x forward primer (20 µM), and 1.0 µL 10x reverse primer (20 µM), and 2.0 µL of BCR-enriched product. The following PCR cycling conditions were used: for samples generated with Seq-Well, 1 cycle of 95 ℃, for 3 min; 25 cycles of 98 ℃ for 40 s, 67 ℃ for 20 s, 72 ℃ for 1 min; 1 cycle of 72 ℃ for 5 min; for samples generated with 10x Genomics 3'GEX, 1 cycle of 98 ℃, for 2 min; 25 cycles of 98 ℃ for 30 s, 63 ℃ for 20 s, 72 ℃ for 1 min; 1 cycle of 72 ℃ for 5 min. The reactions were then pooled and purified using a homemade SPRI reagent at a ratio of 0.80x[83].

## Construction and sequencing of BCR sequencing libraries

We designed and constructed Nextera-IGHV, Nextera-IGKV, and Nextera-IGLV primers for both human and rhesus macaque. For rhesus macaque, we used modified sets primers adapted from ref. 67. For human and rhesus macaque, heavy chain primers were based on the BIOMED-2 primer system[68]. For human light chain, primers were adapted from ref. 20. (U6 spike-in experiment) or Jiang et al. (PBMC data in Figs. 2 and 3)[84] (Supplementary Data 3). Reaction mixtures for heavy and light chains were

composed of: 12.5 μL 2x Kapa Hifi Hotstart PCR Readymix, 6.0 μL water, 2.5 μL primer mix, and 4.0 μL of enriched product. Primer extension was performed with the following thermal program: 98 °C, 5 min, 55 °C, 30 s, 72 °C, 2 min. The products were then cleaned with SPRI at a ratio of 0.8x (light chain) or 0.65x (heavy chain) and were eluted into 11 μL of water.

Four reactions of library PCR were performed per sample. Reactions were composed of: 0.5 μL P5-Seq-Well primer (10 μM), 0.5 μL P7-Nextera primer (10 μM), 12.5 μL 2x Kapa Hifi Hotstart Readymix, 9 μL of water, 2.5 μL of primer extension product for samples generated with Seq-Well; 0.5 μL P5-TruSeqR1 primer (10 μM), 0.5 μL P7-Nextera primer (10 μM), 12.5 μL 2x Kapa Hifi Hotstart Readymix, 9 μL of water, 2.5 μL of primer extension product for samples generated with 10×3'GEX. Amplification used the following cycling conditions: 1 cycle, 95 °C, 2 min; 14–20 cycles of 95 °C, 30 s, 60 °C, 30 s, 72 °C, 1.5 min; 1 cycle of 72 °C, 5 min. Reactions were pooled and purified using SPRI at a ratio of 0.65x (heavy chain) or 0.8x (light chain). Final products were assessed using an Agilent Tapestation. Light chain libraries usually show a clean peak around 1000 bp, and heavy chain libraries usually show a clean peak around 1600 bp.

Libraries were sequenced on an Illumina Miseq using 600 cycle kits or on an Illumina Novaseq 600 using 500 cycle kits. For samples prepared with Seq-Well, the Seq-Well R1 primer was used to sequence the cell barcode and UMI (20 bp on Miseq, 26 bp on Novaseq). Then, custom sequencing primers, specific for the BCR constant region, were used to sequence the BCR using the index 1 read (300 bp on Miseq, 270 bp on Novaseq) (Supplementary Data 3). The 8-base pair-length i5 index barcode was sequenced using the Seq-Well R2 index primer (Novaseq) or no custom primer (Miseq). Lastly, the Nextera primer was used to sequence the remainder of the BCR with read 2 (300 bp on Miseq, 220 bp on Novaseq). For samples prepared with 10x Genomics 3'GEX, the Mod-TruSeqR1 primer was used to sequence the cell barcode and UMI (20 bp on Miseq, 26 bp on Novaseq). Then, custom sequencing primers, specific for the BCR constant region, were used to sequence the BCR using the index 1 read (300 bp on Miseq, 270 bp on Novaseq). The 8-base pair-length i5 index barcode was sequenced using the Mod-TruSeq-Index2 primer (Novaseq) or no custom primer (Miseq) (Supplementary Data 3). Lastly, the Nextera primer was used to sequence the remainder of the BCR with read 2 (300 bp on Miseq, 220 bp on Novaseq). Based on our analysis of sequencing depth (Supplementary Fig. 2c–e), we recommend sequencing these libraries to a depth of 2000 to 20,000 reads per B cell in the original sample.

## Processing of raw single-cell BCR sequencing data

Processing of BCR sequence data was performed with pRESTO and Change-O from the Immcantation software suite[85,86]. First, potential errors in cell barcode and UMI sequence were corrected for errors up to one nucleotide mismatch with a directional UMI collapse, as implemented in the UMI-Tools[87]. Data was filtered by Q-score using the FilterSeq.py function to remove any sequences with an average Q score below 25. Then, the MaskPrimers.py function was used to annotate sequences with the correct isotype (R1 index) and V-region (R2) primer, as well as to mask the corresponding primer regions. The PairSeq.py function was used to retain only read pairs that passed both the FilterSeq.py and MaskPrimers.py processing steps. The data was then segregated by molecular identity (cell barcode + UMI). For samples generated with Seq-Well, the BuildConsensus.py function was used to call consensus sequences for read 1 index and read 2 of each unique molecule (cell barcode + UMI) separately. For samples generated with 10x Genomics 3'GEX, the ClusterSet.py function was first used to subcluster sequence reads sharing the same molecular identity (cell barcode + UMI), the fraction of each subcluster was calculated, and only the subcluster with more than half of the total read count of each molecular identity were kept; lastly, the BuildConsensus.py function was used to call consensus sequences for read 1 index and read 2 of each unique molecule. The AssemblePairs.py function was used to assemble these consensus sequences into a single overlapping sequence. The resulting sequences were then analyzed with IgBlast[88] using reference sequences provided by IMGT.

## Downstream processing of single-cell BCR data

Molecular consensus sequences were filtered based on the total number of reads (separate for each dataset): >5 reads: rhesus macaque data; >10 reads: PBMC data, heavy chain for HEK experiment; >20 reads: light chain for HEK experiment. Molecular consensus sequences that contained greater than four ambiguous "N" characters were also discarded. BCR sequences were matched to single-cells using the single-cell barcodes with a 1 nucleotide Hamming distance error tolerance. To determine a single consensus sequence for each cell, we first considered all sequences attributed to a single cell. If multiple sequences were recovered, we determined a cell-level consensus sequence as follows: first, we performed single-linkage clustering on the recovered IMGT-gapped sequences using Levenstein distance and cut the resulting dendrogram at a height of five to produce clusters of closely related sequences. We further considered sequences that belonged to the largest cluster and had uniform lengths and determined a gapped consensus sequence. If the resulting consensus retains ambiguous ("N") characters, we discarded the sequence that was supported by the fewest number of reads and re-attempted to determine a consensus sequence. This process allowed us to combine information from multiple BCR molecules recovered from the same cell into a single cell-level consensus sequence.

## Analysis of B cell clonotypes

Clonotypes were determined using the DefineClones.py function in Change-o[86], using a Hamming distance calculation and a similarity threshold of 0.10. Germline sequences were then constructed using the CreateGermlines.py function, and frequencies of SHM were determined using the observedMutations function, implemented in Shazam, using the predicted germline sequence with a masked D gene.

## Bulk BCR sequencing

Bulk heavy chain variable region sequencing was performed using 5′-RACE. RNA from magnetically isolated B cells was extracted using the Nucleospin XS RNA kit (Machery-Nagel), eluted into 12 μL of RNAse-free water, and stored at −80 °C until use. To perform reverse transcription (RT), 6 μL of RNA was combined with 2 μL of primer mix containing 20 μM each of RT primers for IgM, IgG, IgA, IgK, and IgL and 400 U/uL NxGen RNAse inhibitor (LGC Biosearch). This mixture was then heated to 70 °C for 4 min, cooled to 42° for 3 min, and cooled to 4 °C. The RT reaction mixture was prepared on ice and contained (per sample): 4 μL SmartScribe 1st Strand 5X buffer (Takara), 2 μL 100 μM DTT (Takara), 2 μL 10 μM template switching oligo, 2 μL 10 mM dNTPs (New England Biosciences), and 2 μL Smartscribe Reverse Transcriptase (Takara). RNA mixtures were combined with the RT reaction mixture, and RT was performed at 42 °C for 90 min, followed by heating to 70 °C for 10 min and cooling to 4 °C. PCR1 was performed separately for light and heavy chains; the reaction mixture was then prepared as follows: 5 μL RT product, 5 μL 5X Phusion Buffer (NEB), 10 μL 1 M Trehalose (Life Sciences Advanced Technologies), 2.5 μL 10 μM 5′-PCR1 primer, 2.5 μL each PCR1 3′-constant region primer, 1 μL Phusion polymerase (NEB), and remainder water (to 50 μL). PCR1 was performed as: 1 cycle, 98 °C, 1 min; 25 cycles: 98 °C for 20 s, 58 °C for 30 s, 72 °C for 1 min; 1 cycle, 72 °C for 1 min, hold at 4 °C. PCR2 reaction mixture was performed as follows: 18 μL water, 25 μL 2X Kapa HiFi HotStart Readymix (Roche), 1.5 μL 5′-PCR2 primer (10 μM), 1.5 μL 3′-PCR2 primer (10 μM). PCR2 was performed as follows: 14–18 cycles of: 98 °C for 40 s, 65 °C for 20 s, 72 °C for 30 s; 1 cycle of 72 °C for 5 min, hold at 4 °C. Library quality was assessed using an Agilent Tapestation D5000 assay, and libraries were sequenced on an Illumina Novaseq SP 500 cycle kit, using 250 × 250 bp reads. All primers were purchased from Integrated DNA Technologies (IDT). Primer sequences used for bulk BCR sequencing are contained in Supplementary Data 4.

## Processing of bulk BCR data

Processing of bulk BCR sequence data was performed with pRESTO and Change-O from the Immcantation software suite[85,86]. First, the FilterSeq.py function was used to remove reads with an average Q-score of less than 25.

Then, MaskPrimers.py was used to identify the 14 base pair-long UMI appended during RT as well as the constant region isotype. The PairSeq.py function was used to retain only read pairs that passed both the FilterSeq.py and MaskPrimers.py processing steps. Consensus sequences for each UMI were then assembled using the BuildConsensus.py function, and the AssemblePairs.py function was used to assemble read mates into a single BCR, using sequential-guided assembly with an IMGT reference. The resulting sequences were then analyzed with IgBlast[88] using reference sequences provided by IMGT. For comparison to single-cell BCR data, we used only unique, functional sequences supported by greater than 5 reads.

### Processing of cell hashing data

Cell hashing data was aligned to HTO barcodes using CITE-seq-Count v1.4.2. To establish thresholds for positivity for each HTO barcode, we first performed centered log-ratio normalization of the HTO matrix and then performed k-medioids clustering with $k = 5$ (one for each HTO). This produced consistently five clusters, each dominated by one of the 5 barcodes. For each cluster, we first identified the HTO barcode that was dominant in that cluster. We then considered the threshold to be the lowest value for that HTO barcode among the cells classified in that cluster. To account for the scenario in which this value was substantially lower than the rest of the values in the cluster, we used Grubbs' test to determine whether this threshold was statistically an outlier relative to the rest of the cluster. If the lower bound was determined to be an outlier at $p = 0.05$, it was removed from the cluster, and the next lowest value was used as the new threshold. This procedure was iteratively applied until the lowest value in the cluster was no longer considered an outlier at $p = 0.05$. Cells were then determined to be "positive" or "negative" for each HTO barcode based on these thresholds. Cells that were positive for multiple HTOs or were negative for all HTOs were excluded from downstream analysis. To account for differences in sequencing depth between samples, these steps were performed separately for each Seq-Well array that was processed.

### Reanalysis of published datasets

Three independent studies that performed BCR recovery from samples of human B cells isolated from PBMC or lymphoid tissues (tonsils) using 10x Genomics 5′ Immune Profiling were selected to benchmark the sensitivity of our method[25,28,52]. The processed and annotated Seurat object from King et al. was downloaded from ArrayExpress with the accession number E-MTAB-9005. Processed data from Dugan et al. were retrieved from GEO with the accession number GSE171703. Processed data from Zurbuchen et al. were downloaded from Zenodo at https://doi.org/10.5281/zenodo.7064118. Data were processed according to the descriptions in their respective method sections. The recoveries of heavy chain, light chain, and paired chains were calculated for each reaction lane.

### Antibody expression and Luminex assay

BCR sequences were selected from the top twenty-four most expanded clones for antibody expression (Supplementary Data 2) by the TurboCHO-HT 2.0 service (GenScript). Antibodies were expressed as Rhesus-murine chimeras in murine IgG1, kappa, and lambda backbones with the signal peptide MGWSCIILFLVATATGVHS. Purified antibodies by size exclusion high-performance liquid chromatography (SEC-HPLC) were stored in PBS buffer (pH 7.4).

Antibody specificity was evaluated by 96-plex Luminex-based direct immunoassay (dLIA) using antigen-coupled MagPlex beads according to the manufacturer's protocol. Briefly, antigens (25 μg/mL in 20 mM HEPES, 5 mL) and methylated human serum albumin (50 μg/mL in 20 mM HEPES, 5 mL) were pooled in light protective tubes and incubated at room temp for 1 h with gentle mixing. MagPlex beads ($1.25 \times 10^7$ beads/mL, 1 mL) were added to the reaction tubes and incubated at 4 °C for 18 h with gentle mixing. A magnetic tube stand was used to pellet beads for washing with storage solution (1% BSA and 0.05% $NaN_3$ in 0.01 M DPBS, $3 \times 10$ mL). Antigen-coupled beads in storage solution (10 mL) were stored at 4 °C

before use. Antigen-coupled MagPlex beads were blocked with 0.5% BSA in EIA-7 buffer for 1 h in light protective tubes and transferred to black 96-well microplates (200,000 beads/region). Plates were washed with EIA-7 ($4 \times 150$ μL) followed by the addition of purified antibodies (0.1–12,500 ng/mL, 100 μL/well, 500 rpm for 1 h). Plates were washed with EIA-7 ($4 \times 150$ μL) followed by the addition of R-Phycoerythrin AffiniPure™ Goat Anti-Mouse IgG (subclasses 1 + 2a + 2b + 3), Fcγ Fragment Specific (Jackson ImmunoResearch Inc) (1 μg/mL, 100 μL/well, 500 rpm for 1 h). Plates were washed with EIA-7 ($4 \times 150$ μL per well) and beads were resuspended in EIA-7 (120 μL/well, 500 rpm for 5 min). Net mean fluorescent intensities (MFI) were measured with a Bio-Plex 3D suspension array system via xPonent software (Version 4.3.309.1).

### Statistics and reproducibility

Statistical tests and $p$ values were performed and calculated in R. The choices of the statistical tests were stated in the figure legends. The alpha level was 0.05. Two-tailed tests were used unless specifically stated. For Figs. 2 and 3, two vials of PBMCs were profiled by scRNA-seq using Seq-Well and 10x Genomics 3′ GEX, respectively. Each vial of PBMC was split into eight replicates, wherein four of the eight replicates were enriched for B cells by magnetic isolation, and loaded onto Seq-Well arrays or 10x Genomics Chromium reaction lanes, respectively. For Fig. 4, the mixture of PBMCs and HEK cells was split into six replicates and loaded onto Seq-Well arrays.

### Reporting summary

Further information on research design is available in the Nature Portfolio Reporting Summary linked to this article.

### Data availability

Raw data generated with Seq-Well from this study has been deposited on GEO with accession number GSE232873. Raw data generated with 10x Genomics 3′GEX from this study has been deposited on GEO with accession number GSE266697. The U6 BCR plasmids have been deposited on Addgene (#223845, #223846). All other data are available upon reasonable request from the corresponding author Dr. J. Christopher Love (clove@mit.edu).

### Code availability

The codes used to analyze BCR data and produce figures in this manuscript are available at on GitHub https://github.com/duncanmorgan/BCR_Recovery_3prime_scRNAseq or Zenodo (https://doi.org/10.5281/zenodo.12735398)[89]. The following software were used: Python v3.6.4, pRESTO v0.5.6, Change-O v0.4.6, IgBLAST v1.14.0, R v4.1.1, Seurat v4.1.1, FlowJo v10.8.1, FastxToolKit v0.0.13, ggplot2 v3.5.0, dplyr v1.1.4, pheatmap v1.0.12, rcolorbrewer v1.1.3, ggrastr v1.0.2, ggbeeswarm v0.7.2, stringr v1.5.1, and reshape2 v1.4.4. Scripts for processing all raw sequencing data and generating all figures will be made available upon reasonable request.

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

## Acknowledgements

We thank N. Kamelamela, C. Hallee, S. Levine, and the MIT BioMicro Center for assistance with library preparation and sequencing, as well as the KI Flow Cytometry Core and the New Iberia Research Center (NIRC). We thank Christian Guerrero and Hal Jones for helping with the Luminex assay. This work was supported in part by the Koch Institute Support (core) NIH Grant P30-CA14051 from the National Cancer Institute, as well as the Koch Institute – Dana-Farber/Harvard Cancer Center Bridge Project. This work was also supported by Pfizer Incorporated. S.U.P. received support from the NIH (5R01AI155630), the Charles H. Hood Foundation Child Health Research Award, and the Food Allergy Science Initiative.

## Author contributions

Conceptualization: D.M.M., J.C.L., L.C., I.K. Methodology: D.M.M., Y.J.Z., J.K., S.T.S., M.M., S.S., J.L., N.S., S.U.P. Investigation: D.M.M., Y.J.Z., J.K., M.M., S.S., J.L., N.S., S.U.P. Visualization: D.M.M., Y.J.Z., O.S., E.F. Funding acquisition: J.C.L., L.C., I.K., D.J.I. Project administration: J.C.L., I.K., L.C. Supervision: J.C.L., I.K., L.C. Writing – original draft: D.M.M., Y.J.Z., J.C.L. Writing – review & editing: D.M.M., Y.J.Z., D.J.I., J.C.L., L.M.

## Competing interests

The authors declare the following competing interests: J.C.L. has interests in Sunflower Therapeutics PBC, Honeycomb Biotechnologies, OneCyte Biotechnologies, SQZ Biotech, Alloy Therapeutics, QuantumCyte, Amgen, and Repligen (these interests are reviewed and managed under Massachusetts Institute of Technology's policies for potential conflicts of interest); in addition, he receives sponsored research support at Massachusetts Institute of Technology from Amgen, the Bill and Melinda Gates Foundation, Biogen,

Pfizer, Sartorius, Mott Corp, TurtleTree, Takeda, and Sanofi, and his spouse is an employee of Sunflower Therapeutics PBC. J.K., M.M., S.S., L.J., N.S., I.K., and L.C. are employees of Pfizer and may, as a consequence, be shareholders. Pfizer was involved in the design, analysis, and interpretation of the data in this research study, the writing of this report, and the decision to publish. SUP consults for Mabylon and Buhlmann and conducts a clinical trial for Regeneron. The other authors declare no competing interests.
