## [Peer Review File · Communications Biology]

Reviewers' comments:

Reviewer #1 (Remarks to the Author):

Morgan et al. developed an oligo-enriched method to perform BCR analysis for 3'-barcoded single-cell cDNA. The authors utilized the enriched B cells from the PBMC samples and the HEK cells specifically expressed U6 BCR clonotype to assess the reliability and the sensitivity of the method for BCR paired analysis while compared to the information of BCR clonotypes extracted from the single-cell transcriptome or the coverage/complexity of the BCR heavy or light sequences revealed by bulk BCR V(D)J sequencing. Lastly, Morgan et al. applied the method to explore and identify few antigen-reactive BCR clonotypes in the NHP individuals that were vaccinated or unvaccinated with ST3 antigen.

Overall, the content of the manuscript is pleasant to read and concise but understandable. However, there are few typos or mild errors needed to be fixed or modified in the sections of Materials & Methods or figures. In addition, I have few comments or suggestions for the authors. Hopefully, what I shared below could be helpful for the authors to publish the method, especially in the application of recovery of the BCR information from the 3'-barcoded cDNA of the B cells or related samples in the freezers of lots of the researchers in this field, with more comprehensive information and meanwhile being suitable for publication in Communication Biology.

Major comments:

1. In introduction, the authors claimed this method should be able to apply to most of the commercial 3'-barcoded cDNA chemistries. However, only Seq-Well platform was mentioned and used for data generation in this manuscript. To provide more scientific impact to the field, I would like to recommend the author also apply their method on the different 3' chemistry or cDNA reagents (e.g. 10x Genomics 3' cDNA or any drop-seq cDNA) to generate the BCR data and compare it with the Seq-well one.
2. Although the authors performed sensitivity analysis based on the levels or the counts of the BCR transcripts via Seq-well scRNAseq, the author should benchmark the feasibility and the sensitivity of their method comparative to at least one of the widely-used single-cell VDJ sequencing methods like 10x Genomics 5' VDJ chemistry.
3. The authors found the shared BCR clonotypes from ST3-reactive B cell in either vaccinated or unvaccinated animals. However, the authors did not provide direct evidence to support they, at least most or some of them if not all, are ST3 antigen specific. Hope the authors can perform a functional assay to support the antibody produced by using the sequences that were revealed by this method can recognize ST3 antigen specifically (e.g. ELISA analysis)

Minor comments:

1. There is an inconsistent way to symbolize the temperature degree in multiple Materials &

Methods (M&M) sections or paragraphs. Please specify and correct it as oC not C.

2. The section of single-cell RNA-sequencing in M&M was written based on CD4 memory T cell. Shall it be rewritten for “B cell” sequencing?

3. In the section of HCK cell culture and transfection in M&M, “....culture at 378C....” may be able to delete

4. In the section of Construction and sequencing of BCR sequencing libraries in M&M, would “...., 4.08 uL primer mix....” Be the typo of “....2.5 uL....”?

5. The A-to D steps in figure 1 did not address or state appropriately in the figure legend.

6. In figure 6B, I think you missed a “ Fos-activated” sample label?

7. The observation of the U6 BCR sequencing in the HEK cells suggested the starting input (the amount of the Seq-well cDNA in term of cell numbers) or the PCR cycles may be play role in accuracy of the BCR sequences under a given amount of oligos for pull down. Could the author elaborate the outcome and in the same time provide evidence(s) if there is a way to reduce the PCR or sequencing errors by fine tuning the input amount and/or changing the PCR cycles?

8. In Figure 2 G, the platform seemed to be more sensitive to IGL detection than WTA. However, when you look carefully into figure 2 E and F, this method enhanced some of the low-expressed IGH sequences but significantly attenuated the sensitivity of detection for a good portion of IGH sequences Interestingly, the case of IgK is kind of both. Do you have any explanation for such variation of detection sensitivity between isotypes? Can you discuss and address more about how kind of the bias in BCR detection may occur by using this method?

9. The cDNA is 3'-barcoded with UMIs. Could you provide and discuss about the sequencing qc metrics related to UMI detection and sequencing coverage saturation? How is the read coverage redundancy of each BCR counts shown in the dataset of your samples? Could tell us more about how you use UMI to calibrate or quantify your WTA or BCR sequencing raw data?

10. In Figure 3B, you observed a good amount of the cells in individual isotype category also detected the other BCR isotypes (mainly IgM across the isotype bins). Can you kindly provide any explanation or data to show or proof it results from partial co expression during the transition of cell sates or artifact results due to doublet or multiplet per well, non-specific pulldown, or ambient RNA contamination?

Reviewer #2 (Remarks to the Author):

Morgan and colleagues described a 3'-based approach to BCR sc-seq. Such an approach is of high interest and usefulness to the field.

Major

- Have you compared 5' with 3' libraries from the same sample? To check if there are any differences between the two approaches? Can you discuss this?
- To what extent can the technology help resolve germline gene polymorphism?
- Ig isotypes: can you also differentiate by subtype?

Minor

- Can you provide in Figure 1 a comparison of 3' and 5' advantages? I think this would help the reader understand the advances made in this paper.

Reviewer #3 (Remarks to the Author):

In this manuscript, Morgan et al describe a technically adept method for extracting V(D)J sequences from 3' scRNAseq libraries without losing the barcoding information. Overall, their technique is well-validated, and the authors use it for an integrated analysis of the transcriptomic phenotypes and BCR sequences of vaccine responses to ST3 in macaques. All of the work is of high quality and is more than suitable for publication.

My only complaint is that, from the brief dismissal of "5'-barcoded library constructions" in the introduction, a reader wouldn't know that it's actually a wide-spread -standard, even- technique with a thriving commercial ecosystem behind it. Two recent examples, off the top of my head:

<https://www.nature.com/articles/s41392-021-00610-7>

<https://www.nature.com/articles/s41590-021-01088-9>

The authors presumably have reasons for favoring 3' library construction that have led them to develop this alternative technique, but I would like to see them actually describe that thought process.

Morgan et al. developed an oligo-enriched method to perform BCR analysis for 3'-barcoded single-cell cDNA. The authors utilized the enriched B cells from the PBMC samples and the HEK cells specifically expressed U6 BCR clonotype to assess the reliability and the sensitivity of the method for BCR paired analysis while compared to the information of BCR clonotypes extracted from the single-cell transcriptome or the coverage/complexity of the BCR heavy or light sequences revealed by bulk BCR V(D)J sequencing. Lastly, Morgan et al. applied the method to explore and identify few antigen-reactive BCR clonotypes in the NHP individuals that were vaccinated or unvaccinated with ST3 antigen.

Overall, the content of the manuscript is pleasant to read and concise but understandable. However, there are few typos or mild errors needed to be fixed or modified in the sections of Materials & Methods or figures. In addition, I have few comments or suggestions for the authors. Hopefully, what I shared below could be helpful for the authors to publish the method, especially in the application of recovery of the BCR information from the 3'-barcoded cDNA of the B cells or related samples in the freezers of lots of the researchers in this field, with more comprehensive information and meanwhile being suitable for publication in Communication Biology.

Major comments:

1. In introduction, the authors claimed this method should be able to apply to most of the commercial 3'-barcoded cDNA chemistries. However, only Seq-Well platform was mentioned and used for data generation in this manuscript. To provide more scientific impact to the field, I would like to recommend the author also apply their method on the different 3' chemistry or cDNA reagents (e.g. 10x Genomics 3' cDNA or any drop-seq cDNA) to generate the BCR data and compare it with the Seq-well one.
2. Although the authors performed sensitivity analysis based on the levels or the counts of the BCR transcripts via Seq-well scRNAseq, the author should benchmark the feasibility and the sensitivity of their method comparative to at least one of the widely-used single-cell VDJ sequencing methods like 10x Genomics 5' VDJ chemistry.
3. The authors found the shared BCR clonotypes from ST3-reactive B cell in either vaccinated or unvaccinated animals. However, the authors did not provide direct evidence to support they, at least most or some of them if not all, are ST3 antigen specific. Hope the authors can perform a functional assay to support the antibody produced by using the sequences that were revealed by this method can recognize ST3 antigen specifically (e.g. ELISA analysis)

Minor comments:

1. There is an inconsistent way to symbolize the temperature degree in multiple Materials & Methods (M&M) sections or paragraphs. Please specify and correct it as °C not C.
2. The section of single-cell RNA-sequencing in M&M was written based on CD4 memory T cell. Shall it be rewritten for "B cell" sequencing?
3. In the section of HCK cell culture and transfection in M&M, "...culture at 378C...." may be able to delete
4. In the section of Construction and sequencing of BCR sequencing libraries in M&M, would "..., 4.08 uL primer mix...." Be the typo of "...2.5 uL...."?
5. The A-to D steps in figure 1 did not address or state appropriately in the figure legend.

6. In figure 6B, I think you missed a “ Fos-activated” sample label?
7. The observation of the U6 BCR sequencing in the HEK cells suggested the starting input (Seq-well cDNA in term of cell numbers) or the PCR cycles may be play role in accuracy of the BCR sequences under a given amount of oligos for pull down. Could the author elaborate the outcome and provide evidence(s) if there is a way to reduce the PCR or sequencing errors by fine tuning the input amount and/or changing the PCR cycles?
8. In Figure 2 G, the platform seemed to be more sensitive to IGL detection than WTA. However, when you look carefully into figure 2 E and F, this method enhanced some of the low-expressed IGH sequences but significantly attenuated the sensitivity of detection for a good portion of IGH sequences Interestingly, the case of IgK is kind of both. Do you have any explanation for such variation of detection sensitivity between isotypes? Can you discuss and address more about how kind of the bias in BCR detection may occur by using this method?
9. The cDNA is 3'-barcoded with UMIs. Could you provide and discuss about the sequencing qc metrics related to UMI detection and sequencing coverage saturation? How is the read coverage redundancy of each BCR counts shown in the dataset of your samples? Could tell us more about how you use UMI to calibrate or quantify your WTA or BCR sequencing raw data?
10. In Figure 3B, you observed a good amount of the cells in individual isotype category also detected the other BCR isotypes (mainly IgM across the isotype bins). Can you kindly provide any explanation or data to show or proof it results from partial co expression during the transition of cell sates or artifact results due to doublet or multiplet per well, non-specific pulldown, or ambient RNA contamination?

Reviewer #1 (Remarks to the Author):

Morgan et al. developed an oligo-enriched method to perform BCR analysis for 3'-barcoded single-cell cDNA. The authors utilized the enriched B cells from the PBMC samples and the HEK cells specifically expressed U6 BCR clonotype to assess the reliability and the sensitivity of the method for BCR paired analysis while compared to the information of BCR clonotypes extracted from the single-cell transcriptome or the coverage/complexity of the BCR heavy or light sequences revealed by bulk BCR V(D)J sequencing. Lastly, Morgan et al. applied the method to explore and identify few antigen-reactive BCR clonotypes in the NHP individuals that were vaccinated or unvaccinated with ST3 antigen.

Overall, the content of the manuscript is pleasant to read and concise but understandable. However, there are few typos or mild errors needed to be fixed or modified in the sections of Materials & Methods or figures. In addition, I have few comments or suggestions for the authors. Hopefully, what I shared below could be helpful for the authors to publish the method, especially in the application of recovery of the BCR information from the 3'-barcoded cDNA of the B cells or related samples in the freezers of lots of the researchers in this field, with more comprehensive information and meanwhile being suitable for publication in Communication Biology.

We appreciate the reviewer's comments about our manuscript and the usefulness of our approach.

Major comments:

1. In introduction, the authors claimed this method should be able to apply to most of the commercial 3'-barcoded cDNA chemistries. However, only Seq-Well platform was mentioned and used for data generation in this manuscript. To provide more scientific impact to the field, I would like to recommend the author also apply their method on the different 3' chemistry or cDNA reagents (e.g. 10x Genomics 3' cDNA or any drop-seq cDNA) to generate the BCR data and compare it with the Seq-well one.

We thank the reviewer for the suggestion. We have now applied our approach to recover BCR sequences from a cDNA library generated using the 10x Genomics 3' GEX platform. We show that the recovery of BCR sequences compares well to that for Seq-Well, and that these sequences exhibit similar quality control metrics. These data indicate that our method can extend to other 3'-barcoded single-cell platforms as well. We have incorporated these new results into our manuscript, primarily to the amended Figure 2 and Figure 3, which appear below:

Figure 2. Recovery of full-length, paired BCR sequence from single-cell libraries. **(a)** UMAP of cell phenotypes present in human PBMC prepared using Seq-Well ($n = 24,806$ cells). UMAP of BCR recovery from single cells prepared using Seq-Well. **(c)** Fraction of cells with no recovery, recovery of heavy chain, recovery of light chain, and paired recovery using Seq-Well

(n = 8 samples). Error bars are mean +/- standard error of the mean **(d)** UMAP of cell phenotypes present in human PBMC prepared using 10x Genomics 3` v3 (n = 50,877 cells). **(e)** UMAP of BCR recovery from single cells prepared using 10x Genomics 3` v3. **(f)** Fraction of cells with no recovery, recovery of heavy chain, recovery of light chain, and paired recovery using 10x Genomics 3` v3 (n = 8 samples). Error bars are mean +/- standard error of the mean. **(f-i)** Correlation between the number of counts mapping to the IGH/IGK/IGL locus and the number of functional heavy chain or light chain molecules recovered from Seq-Well libraries. Spearman's correlation coefficient and the associated p-value are shown. **(j-l)** Correlation between the number of counts mapping to the IGH/IGK/IGL locus and the number of functional heavy chain or light chain molecules recovered from 10x Genomics libraries. Spearman's correlation coefficient and the associated p-value are shown.

Figure 3. Concordance between single-cell whole transcriptome and BCR libraries. **(a, b)** UMAP of isotypes of BCR heavy chain sequences recovered from single-cell libraries prepared by Seq-Well (a) and 10x Genomics 3' GEX (b), respectively. **(c, d)** Distribution of recovered BCR isotypes among B cells binned by the most common Ig constant region transcript present in WT. (c) is for Seq-Well-prepared samples, and (d) for 10x Genomics 3' GEX. **(e, f)** Isotypes of BCR heavy chain sequences recovered from single-cell libraries, grouped by phenotypes assigned in single-cell gene expression data from Seq-Well (e) and 10x Genomics 3' GEX (f),

respectively. All genes shown have at least two transcripts recovered in both WTA and BCR libraries. **(g, h)** Heat map comparing most common heavy chain V gene segments in WT libraries and V-genes of recovered BCR sequences. **(i, j)** Heavy chain somatic mutation frequency overlaid onto UMAP of cells prepared by Seq-Well (i) and 10x Genomics 3' GEX (j), respectively. **(k, l)** Somatic mutation frequency of heavy chain BCR sequences grouped by B cell phenotypes. **(m, n)** Somatic mutation frequency of light chain BCR sequences grouped by B cell phenotypes. (k, m) are for Seq-Well-prepared samples, and (l, n) for 10x Genomics 3' GEX. **(o, p)** Somatic mutation frequency of BCR sequences grouped by B cell isotypes. P-values are calculated using a two-sided Wilcoxon rank-sum test and are adjusted using Bonferroni correction. ns $p > 0.05$, * $p \leq 0.05$, ** $p \leq 0.01$, *** $p \leq 0.001$, **** $p \leq 0.0001$.

2. Although the authors performed sensitivity analysis based on the levels or the counts of the BCR transcripts via Seq-well scRNAseq, the author should benchmark the feasibility and the sensitivity of their method comparative to at least one of the widely-used single-cell VDJ sequencing methods like 10x Genomics 5' VDJ chemistry.

We acknowledge and appreciate this suggestion. We have incorporated a reanalysis of three previously published datasets of human B cells analyzed with 10x Genomics 5'-VDJ chemistry¹⁻³. Across these datasets, in comparison with the datasets we have generated using our method for BCR recovery from 3'-barcoded libraries, we find that the studies based on recovery from 5'-barcoded libraries achieve marginally higher rates of recovery per B cell analyzed.

We note that there can be several factors that affect recovery, however, including the phenotypes and quality of input cells (e.g., plasma cells, plasmablasts, resting memory B cells), RNA capture efficiency, and data pre-processing that can make these direct data comparisons more difficult to interpret. To attempt to normalize this analysis and focus on comparing the features related to the innovation here (improved recovery of BCRs from samples), we have also compared the rates of BCR recovery on a molecular basis, rather than a cellular basis. In this analysis, we find that the performance of our method relative to published 5'-barcoded datasets is quite similar for heavy chain and slightly better for light chain. These analyses together suggest that at least part of the difference in recovery rates observed results from sample-intrinsic qualities and suggests that our method possesses a similar sensitivity for BCR recovery as these commercially-available platforms.

We have incorporated this analysis as Figure 5 in our manuscript as follows:

LINE 225: To benchmark the sensitivity of our method against commercially available platforms for the recovery of BCR sequences from 5'-barcoded single-cell libraries, we reanalyzed data generated using 5'-barcoded scRNA-seq (10x Genomics) from three independent studies that performed BCR recovery from samples of human B cells isolated from PBMC or lymphoid tissues (tonsils)¹⁻³. We first determined the fraction of B cells with recovery of heavy, light, and paired heavy/light chains in these three datasets of 5'-barcoded libraries with the three datasets of 3'-barcoded libraries we generated with Seq-Well or the 10x Genomics 3'-GEX platform (**Figure 5a**). We found that the recovery obtained with 5'-barcoded libraries (average across all datasets: 70.0% heavy chain, 82.3% light chain, 68.3% paired chains) was only slightly greater than what we obtained with 3'-barcoded libraries (Seq-Well libraries: 67.8% of heavy chain, 57.0% light chain, and 39.9% paired chains; 10x libraries: 56.1% heavy chain, 89.9% light chain, and 52.2% paired chains).

We then compared the sensitivity of these methods on a molecular basis. We determined the relationship between the number of BCR transcripts enumerated in the

WTA product and the probability of BCR recovery from each cell (**Figure 5b**). When adjusted for the number of BCR transcripts enumerated in the WTA product, the recovery of heavy chain sequences was similar between 5'-barcoded and 3'-barcoded libraries; our method for 3'-barcoded libraries slightly outperformed the sensitivity of 5'-barcoded libraries for light chain transcripts. This finding suggests that at least some of the trend towards slightly elevated rates of BCR recovery in the 5'-barcoded libraries examined here result from library-intrinsic features, such as the types of B cells in each sample or improved capture of BCR transcripts during library preparation. Overall, these results confirm that our method for recovery of BCR from 3'-barcoded libraries exhibits comparable sensitivity to commercially available solutions that enable recovery from 5'-barcoded libraries.

Figure 5. Comparison of BCR recovery to 5'-barcoded single-cell platforms. **(a)** Rates of recovery of heavy chain, light chain, and paired chain BCR from either 5'- or 3'-barcoded datasets. Each point represents a single experimental replicate. **(b)** Relationship of the average probability of BCR recovery and the number of heavy chain or light chain transcripts enumerated in whole transcriptome sequencing. Shaded areas represent the standard error of the mean.

3. The authors found the shared BCR clonotypes from ST3-reactive B cell in either vaccinated or unvaccinated animals. However, the authors did not provide direct evidence to support they, at least most or some of them if not all, are ST3 antigen specific. Hope the authors can perform

a functional assay to support the antibody produced by using the sequences that were revealed by this method can recognize ST3 antigen specifically (e.g. ELISA analysis).

We acknowledge and appreciate this suggestion. We generated recombinant antibodies for 24 of the clonotypes identified by our screen, and measured the specificity of these against 96 different pneumococcal serotypes for 23 of these (1 untested due to poor expression yield). The majority of these antibodies showed specificity to Pn3 (the target antigen used to recover the B cells) with 3 showing varying degrees of polyspecificity or no specificity. We have added a description and supporting data to the paper to document these results that affirm the specificity of the clonotypes identified by the sequencing analysis:

Line 349: Lastly, to verify that the BCR motifs identified here were specific for ST3, we selected a total of 24 clonotypes to express as Rhesus-murine IgG1 chimeric antibodies. Of these, 23 expressed and were used to evaluate specificity against 93 different pneumococcal serotypes (**Figure 8a, 8b**). Remarkably, 20 of these 23 clonotypes exhibited strong, specific binding to ST3, and lacked substantial levels of binding to other pneumococcal serotypes. Of the three remaining clonotypes, two exhibited a weak level of polyreactivity to a majority of pneumococcal serotypes evaluated, and one exhibited minimal to no binding to any of the polysaccharides evaluated. We hypothesize that these antibodies may lack reactivity to ST3 in this assay due to structural variations when expression in the murine backbone or due to propagated errors in the original BCR sequence.

Figure 8. Convergent BCR sequences exhibit specific binding to ST3. **(a)** Plots of recombinant Rhesus-murine chimeric antibodies binding to ST3 as a function of concentration and IGHV gene. Each line represents an antibody selected from a single clone. **(b)** Heatmap of antibody binding to 93 different pneumococcal serotypes as a function of IGHV gene family and monkey. The values are shown for MFI measured with 2.3 ug/L of antibody by Luminex. The full list of serotypes evaluated is provided in the Supplemental Table 7.

Minor comments:

1. There is an inconsistent way to symbolize the temperature degree in multiple Materials & Methods (M&M) sections or paragraphs. Please specify and correct it as oC not C.

Thank you. We have corrected this mistake.

2. The section of single-cell RNA-sequencing in M&M was written based on CD4 memory T cell. Shall it be rewritten for “B cell” sequencing?

Thank you. We have corrected this mistake.

3. In the section of HCK cell culture and transfection in M&M, “....culture at 378C....” may be able to delete

Thank you. We have corrected this mistake.

4. In the section of Construction and sequencing of BCR sequencing libraries in M&M, would “...., 4.08 uL primer mix....” Be the typo of “....2.5 uL.....”?

Thank you. We have corrected this mistake.

5. The A-to D steps in figure 1 did not address or state appropriately in the figure legend.

Thank you. We have corrected this mistake.

6. In figure 6B, I think you missed a “ Fos-activated” sample label?

Thank you. We have corrected this mistake.

7. The observation of the U6 BCR sequencing in the HEK cells suggested the starting input (the amount of the Seq-well cDNA in term of cell numbers) or the PCR cycles may be play role in accuracy of the BCR sequences under a given amount of oligos for pull down. Could the author elaborate the outcome and in the same time provide evidence(s) if there is a way to reduce the PCR or sequencing errors by fine tuning the input amount and/or changing the PCR cycles?

We believe that the main determinant of the number of BCR transcripts that we recover using this method is the number of BCR transcripts that undergo successful reverse transcription. Consistent with this hypothesis, we frequently observe strong correlations between the number of BCR transcripts recovered from a single cell in both whole-transcriptome and BCR recovery experiments (**Figure 2g-l, Supplemental Figure 2c-h**).

Into this protocol, we currently load a fixed volume of 3.5 uL, or about 15%, of the total WTA product generated from Seq-Well. This volume retains sufficient WTA product to perform additional replicates of BCR recovery and/or whole-transcriptome sequencing and could not be substantially increased further. The amount of PCR amplification that we perform is the minimum number of cycles that we have observed to consistently generate a BCR amplicon library with sufficient concentration for sequencing (ideally, >2 nM). Loading additional product into the BCR recovery steps could reduce the number of PCR cycles required to generate a sequencing library with this concentration, but the number of PCR cycles would not likely be

substantially reduced (assuming amplification during PCR is 'perfect', a 2x increase in input material would only reduce the number of PCR cycles required by one based on mass input required). Additionally, we would expect that greater amounts of amplification would lead to both increasingly less even coverage of our libraries and the dropout of lowly amplified BCR sequences through mechanisms such as PCR jackpotting⁴ as well as a potential increase in the frequency of PCR-related sequence errors. For these reasons, we have not pursued optimization in this direction presently.

8. In Figure 2 G, the platform seemed to be more sensitive to IGL detection than WTA. However, when you look carefully into figure 2 E and F, this method enhanced some of the low-expressed IGH sequences but significantly attenuated the sensitivity of detection for a good portion of IGH sequences. Interestingly, the case of IgK is kind of both. Do you have any explanation for such variation of detection sensitivity between isotypes? Can you discuss and address more about how kind of the bias in BCR detection may occur by using this method?

We thank the reviewer for raising this question. We initially hypothesized that some degree of this bias may result from variation in the quality of the V gene primers used to amplify these primers on the 5'-end. In our comparison to bulk libraries (generated using 5'-RACE, a library preparation that does not rely on V gene primers; **Figure 4G-H**), we did not find a substantial difference in the overall frequencies of V genes in single-cell libraries recovered from the same sample using 5'-RACE or our method for single-cell BCR recovery. This observation, however, does not rule out that individual BCRs may have acquired mutations in the binding region of these primers that substantially enhance or abrogate the efficiency of the primer extension reaction, causing the efficiency of recovery of individual BCR sequences to be substantially promoted or reduced, relative to other BCRs.

We have expanded on the potential role this phenomenon may play in the discussion of our manuscript.

Line 384: The design of these primers requires the accurate annotation of V genes for a given organism, **as well as an understanding of the frequencies of somatic mutations that are likely to occur that may affect the binding of these primers.**

9. The cDNA is 3'-barcoded with UMIs. Could you provide and discuss about the sequencing qc metrics related to UMI detection and sequencing coverage saturation? How is the read coverage redundancy of each BCR counts shown in the dataset of your samples? Could tell us more about how you use UMI to calibrate or quantify your WTA or BCR sequencing raw data?

We thank the reviewer for raising this point.

As a strategy to analyze our BCR data, we have developed a pipeline in which raw BCR reads are first used to generate a UMI-level consensus sequence. As filtering criteria, we require that each UMI sequence be represented by a minimum number of reads (example: >10) and that the agreement between individual reads is strong, as measured by the number of individual bases for which a consensus can be determined (maximum error: 50%). These filters help to reduce technical noise, such as errors associated with the sequencing instrument and single nucleotides that may accumulate during amplification.

Then, we utilize the consensus sequences from UMI mapping to the same single-cell barcode to generate a cell-level consensus sequence. The agreement between UMI with the same cell barcode is usually strong, but we find the most common form of error is when a minority of sequences appears unrelated to the majority sequence. These unrelated sequences are likely

generated by the formation of chimeric transcripts during PCR or the existence of physical doublets (two B cells in a single well/droplet) in our sequencing data. To address this error, the consensus calling first identifies the largest contingent of related sequences by clustering the BCR sequences of all UMI obtained, and then determines a base-by-base consensus of this majority cluster.

To assess the accuracy of our cell-level consensus sequences, we have performed an analysis in which we compare the sequences of individual UMI sequences with the consensus sequence determined for the corresponding cell. While this analysis lacks truth data, since the true, correct BCR for each cell is unknown, we can consider the consensus sequence determined for each cell to be the most likely BCR. We find that the vast majority of heavy chain and light chain sequences from both Seq-Well and 10x libraries differ from the cellular consensus sequence by less than one nucleotide, supporting a high level of accuracy in individual UMI as well as cellular consensus sequences.

To evaluate the level of sequencing coverage saturation in our BCR libraries, we have evaluated the relationship between the number of transcripts enumerated in whole transcriptome libraries and the number of individual UMI sequences obtained that meet filtering thresholds (>10 sequencing reads). We find that for all of our libraries sequenced, we see a plateau in the rate at which new transcripts are identified as we reach a large number of reads, indicating that these libraries are sequenced to a high degree of saturation. To provide recommendations for how many reads are necessary to obtain a reasonable degree of library saturation, we determined the median number of BCR sequencing reads per B cell recovered in whole-transcriptome sequencing for both Seq-Well and 10x libraries.

Platform	Heavy/light chain	Median number of reads/cell at 90% saturation
10x Genomics 3` v3	Heavy chain	5,610
10x Genomics 3` v3	Light chain	19,852
Seq-Well	Heavy chain	2,379
Seq-Well	Light chain	4,342

We have included this analysis as Supplementary Figure 2 in our manuscript, and we have added recommendations for the sequencing depth of BCR libraries to the methods section of our manuscript.

Line 148: To confirm that our bioinformatic pipelines enabled the accurate reconstruction of BCR sequences, we compared individual UMI sequences recovered from the same cell to the consensus BCR sequence determined by our pipeline. We found strong agreement between individual UMI sequences and cellular consensus sequences in both Seq-Well and 10x libraries, supporting a high degree of sequence accuracy (**Supplemental Figure 3a-b**). A large portion of sequences that did not match the cellular consensus sequence demonstrated little sequence similarity to the cellular consensus sequence (>10 mismatched bases), suggesting that they result from multiple, unrelated BCR transcripts associated with one cell barcode, which may result from physical cell doublets in single-cell droplets or microwells, contamination with ambient BCR sequences, or the generation of chimeric transcripts during PCR amplification. We also assessed the degree of sequencing saturation obtained in these BCR libraries and found that all samples exhibited a plateau in the number of BCR sequences recovered per additional sequencing read, suggesting that additional sequencing depth was unlikely to lead to the recovery of substantially more additional BCR sequences (**Supplemental Figure 3c-e**). Together, these results support that our

method enables the recovery of accurate, full-length BCR sequences from 3'-barcoded scRNA-seq libraries.

Methods:

LINE 593: Based on our analysis of sequencing depth (**Supplemental Figure 3c-e**), we recommend sequencing these libraries to a depth of 2,000 to 20,000 reads per B cell in the original sample.

Supplemental Figure 3. (a, b) Agreement between cellular consensus sequence and individual UMI sequences for all cell barcodes with greater than one UMI recovered from Seq-Well libraries (a) or 10x libraries (b). (c) Library saturation curves, depicting the relationship between the number of sequencing reads per B cell recovered in whole-transcriptome libraries and the number of unique BCR UMI with >10 reads. Each line represents a single technical replicate (Seq-Well array or 10x channel). (d) Number of reads per BCR UMI recovered in whole transcriptome libraries at 90% saturation (90% of unique BCR with >10 reads). (e) Number of reads per BCR UMI recovered in whole transcriptome libraries at 90% saturation (90% of unique BCR with >10 reads).

10. In Figure 3B, you observed a good amount of the cells in individual isotype category also detected the other BCR isotypes (mainly IgM across the isotype bins). Can you kindly provide any explanation or data to show or prove it results from partial co expression during the transition of cell states or artifact results due to doublet or multiplet per well, non-specific pulldown, or ambient RNA contamination?

We thank the reviewer for raising this question.

We can rule out that this contamination results from co-expression during the transition of cell states, because in this case, we would still expect that cells express the same BCR, and we find that this is seldom the case.

We are unable to distinguish between ambient RNA contamination and multiplet cell encapsulation, since both of these sources of error would result in the capture of a full-length RNA that would be enumerated in both whole-transcriptome sequencing and in our BCR recovery method. We do find that if we increase the stringency in which we use whole transcriptome data to assign BCR isotypes by requiring that each cell considered have at least 3 counts of BCR enumerated in its whole transcriptome library, the agreement between isotypes predicted from whole transcriptome sequencing with those recovered using our method appears to increase. This result could suggest that some of the observed errors result from whole-transcriptome sequencing, though sources such as ambient RNA contamination and multiplet cell encapsulation may still contribute.

This adjustment has been incorporated into the updated version of Figure 3.

Reviewer #2 (Remarks to the Author):

Morgan and colleagues described a 3'-based approach to BCR sc-seq. Such an approach is of high interest and usefulness to the field.

We appreciate the reviewer's comments about the interest and usefulness of our approach.

Major

- Have you compared 5' with 3' libraries from the same sample? To check if there are any differences between the two approaches? Can you discuss this?

sequencing methods like 10x Genomics 5' VDJ chemistry.

See also the response to reviewer 1 above.

We acknowledge and appreciate this suggestion. We have now incorporated a reanalysis of three previously published datasets of human B cells analyzed with 10x Genomics 5'-VDJ chemistry¹⁻³. Across these datasets, in comparison with the datasets we have generated using our method for BCR recovery from 3'-barcoded libraries, we find that the studies based on recovery from 5'-barcoded libraries achieve marginally higher rates of recovery per B cell analyzed.

We note that there can be several factors that affect recovery, however, including the phenotypes and quality of input cells (e.g., plasma cells, plasmablasts, resting memory B cells), RNA capture efficiency, and data pre-processing that can make these direct data comparisons more difficult to interpret. To attempt to normalize this analysis and focus on comparing the features related to the innovation here (improved recovery of BCRs from samples), we have also compared the rates of BCR recovery on a molecular basis, rather than a cellular basis. In this analysis, we find that the performance of our method relative to published 5'-barcoded datasets is quite similar for heavy chain and slightly better for light chain. These analyses together suggest that at least part of the difference in recovery rates observed results from sample-intrinsic qualities and suggests that our method possesses a similar sensitivity for BCR recovery as these commercially-available platforms.

We have incorporated this analysis as Figure 5 in our manuscript as follows:

LINE 225: To benchmark the sensitivity of our method against commercially available platforms for the recovery of BCR sequences from 5'-barcoded single-cell libraries, we reanalyzed data generated using 5'-barcoded scRNA-seq (10x Genomics) from three independent studies that performed BCR recovery from samples of human B cells isolated from PBMC or lymphoid tissues (tonsils)¹⁻³. We first determined the fraction of B cells with recovery of heavy, light, and paired heavy/light chains in these three datasets of 5'-barcoded libraries with the three datasets of 3'-barcoded libraries we generated with Seq-Well or the 10x Genomics 3'-GEX platform (**Figure 5a**). We found that the recovery obtained with 5'-barcoded libraries (average across all datasets: 70.0% heavy chain, 82.3% light chain, 68.3% paired chains) was only slightly greater than what we obtained with 3'-barcoded libraries (Seq-Well libraries: 67.8% of heavy chain, 57.0% light chain, and 39.9% paired chains; 10x libraries: 56.1% heavy chain, 89.9% light chain, and 52.2% paired chains).

We then compared the sensitivity of these methods on a molecular basis. We determined the relationship between the number of BCR transcripts enumerated in the WTA product and the probability of BCR recovery from each cell (**Figure 5b**). When adjusted for the number of BCR transcripts enumerated in the WTA product, the recovery of heavy chain sequences was similar between 5'-barcoded and 3'-barcoded libraries; our method for 3'-barcoded libraries slightly outperformed the sensitivity of 5'-barcoded libraries for light chain transcripts. This finding suggests that at least some of the trend towards slightly elevated rates of BCR recovery in the 5'-barcoded libraries examined here result from library-intrinsic features, such as the types of B cells in each sample or improved capture of BCR transcripts during library preparation. Overall, these results confirm that our method for recovery of BCR from 3'-barcoded libraries exhibits comparable sensitivity to commercially available solutions that enable recovery from 5'-barcoded libraries.

Figure 5. Comparison of BCR recovery to 5'-barcoded single-cell platforms. **(a)** Rates of recovery of heavy chain, light chain, and paired chain BCR from either 5'- or 3'-barcoded datasets. Each point represents a single experimental replicate. **(b)** Relationship of the average probability of BCR recovery and the number of heavy chain or light chain transcripts enumerated in whole transcriptome sequencing. Shaded areas represent the standard error of the mean.

- To what extent can the technology help resolve germline gene polymorphism?

An accurate database of germline immunoglobulin alleles is important for many parts of BCR repertoire analysis, including the assignment of clonal lineages and the identification of somatic BCR mutations. In principle, we believe that our method possesses the accuracy to help resolve germline polymorphisms of immunoglobulin genes. We would caution, however, against the application of our method at this time for this purpose, for the following reasons:

- It is often difficult to distinguish novel germline gene polymorphisms from low frequencies of somatic mutations acquired by individual B cells. For this reason, methods such as the genomic sequencing of non-B cells, in which the immunoglobulin loci undergo no genetic alteration, are preferred for this purpose.
- The number of cells that can be analyzed in a single-cell RNA sequencing experiment (1,000s to 10,000s) is often far below the number of observations that would be required to convincingly nominate new allele sequences, especially for rare variants that may constitute <1% of the total immunoglobulin repertoire.

We propose that in situations where a sufficiently accurate or personalized BCR repertoire database is not available, our method could be combined with other methods, including bulk BCR sequencing, to generate a reference dataset of appropriate scale for allele discovery, along with tools such as TiGER, igDiscover, or partis, that allow for the nomination of germline allele repertoires⁵⁻⁷.

To address this point, we have added the following revision to the discussion section of our manuscript:

LINE 388: Nevertheless, the amount of publicly available immunoglobulin repertoire data is increasing, and tools for discovering germline immunoglobulin V genes from repertoire sequencing data have become increasingly more sophisticated, enabling the definition of novel or personalized germline allele databases with greater accuracy and ease. As a result, the characterization of V genes in many species continues to improve, and we expect the availability of suitable primer sets to expand in the future^{6,8,9}.

- Ig isotypes: can you also differentiate by subtype?

The custom sequencing primers used to read the variable region of the BCR in our method are designed to capture between the first 9 to 11 bases of the constant region (the 5'-end of the CH1 exon). This length of sequence is sufficient to differentiate between different isotypes (IgM, IgG, IgA, IgD), but it is generally not enough to differentiate between subtypes (i.e. IgG1 vs. IgG2 or IgA1 vs. IgA2). The majority of sequence variation between subtypes is found in other parts of the constant region, which are not directly sequenced in our approach. We propose, however, that the subtype of antibody produced by an individual B cell could still be identified using read mapping to the constant region in whole transcriptome sequencing, as has been performed in other studies, or could be captured using methods that stain cells using oligonucleotide-labeled antibodies, such as CITE-seq.

We have revised the manuscript to reflect this point:

LINE 99: The custom BCR sequencing primers are designed to capture between 9-11 bases of the constant region, enabling differentiation of BCR constant region isotypes (e.g. IgM vs. IgG). If further determination of BCR subclass is desired (e.g. IgG1 vs. IgG2), this information can be obtained by analyzing the coverage of the BCR constant region obtained in whole-transcriptome sequencing, which is abundant in 3'-barcoded whole transcriptome library preparations¹⁰⁻¹², or additionally captured by protocols that stain cells with oligonucleotide-labeled antibodies, such as CITE-seq¹³.

Minor

- Can you provide in Figure 1 a comparison of 3' and 5' advantages? I think this would help the reader understand the advances made in this paper.

We thank the reviewer for the suggestion. We have now expanded Figure 1 with a corresponding Supplemental Figure 1 to better distinguish how our method differs from those commonly used for 5'-barcoded libraries.

Line 73: In 5'-barcoded single-cell library constructions, amplicons containing the cellular barcode and BCR variable region are generated by the full-length whole-transcriptome amplification (WTA) and subsequent BCR-enrichment with primers specific for the universal primer site (UPS) and the BCR constant regions (**Supplemental Figure 1a, 1b**). The resulting amplicon typically undergoes random fragmentation and further amplification to generate a sequencing library, whereby the sequences of the single-cell barcode and the BCR are simultaneously determined (**Supplemental Figure 1c**).

Supplemental Figure 1. Schematic on the recovery of V(D)J gene sequences using 5'-barcoded scRNA-seq protocol. **(a)** In 5'-barcoded whole-transcriptome sequencing, barcoded cDNA is randomly fragmented and amplified to produce size-defined sequencing libraries. Because the cellular barcode is adjacent to the variable regions of the BCR transcripts, moderate coverage of the variable region is obtained from whole-transcriptome sequencing. **(b)** Additionally, targeted coverage of the BCR variable region can be performed by amplifying 5'-barcoded whole-transcriptome libraries using nested PCR, with outer and inner primers targeting the BCR constant region. These libraries can then be randomly fragmented and sequenced. **(c)** The resulting data allow for in silico reconstruction of the BCR variable region and matching to whole transcriptome libraries by matching cellular barcodes.

Reviewer #3 (Remarks to the Author):

In this manuscript, Morgan et al describe a technically adept method for extracting V(D)J sequences from 3' scRNAseq libraries without losing the barcoding information. Overall, their technique is well-validated, and the authors use it for an integrated analysis of the transcriptomic phenotypes and BCR sequences of vaccine responses to ST3 in macaques. All of the work is of high quality and is more than suitable for publication.

We thank this reviewer for his careful consideration of our work and for the remarks about the quality and suitability of our work for publication.

My only complaint is that, from the brief dismissal of "5'-barcoded library constructions" in the introduction, a reader wouldn't know that it's actually a wide-spread -standard, even- technique with a thriving commercial ecosystem behind it. Two recent examples, off the top of my head: <https://www.nature.com/articles/s41392-021-00610-7>
<https://www.nature.com/articles/s41590-021-01088-9>

The authors presumably have reasons for favoring 3' library construction that have led them to develop this alternative technique, but I would like to see them actually describe that thought process.

We thank the reviewer for raising this point, as it allows us to better highlight the utility of our work. We note that both 3' and 5' barcoded constructions are currently widespread in the academic community. Currently, available commercial assays from 10x Genomics demonstrate similar performance for the measurement of gene expression, the majority of those interested in immunoglobulin repertoire profiling currently opt to utilize 5'-barcoded library constructions, primarily because of a lack of suitable methods for BCR recovery from 3'-barcoded libraries. Despite this use, we note the following advantages for 3'-barcoded library constructions over 5'-barcoded library constructions:

- An increasing number of multi-modal technologies have been developed for 3'-barcoded library constructions but are not available for 5'-barcoded library constructions, including:
 - Single-cell RNA + chromatin accessibility (10x Single-Cell Multiome ATAC + Gene Expression, SNARE-seq, TEA-seq, sci-car)¹⁴⁻¹⁶
 - mRNA + spatial (Slide-Seq V2)^{17,18},
 - mRNA + chromatin accessibility + DNA methylation (snmCAT-seq)¹⁹.

As all of these methods isolate full-length, 3'-barcoded cDNA during library preparation, we believe that our technique for BCR recovery could be adapted for compatibility with any of these methods.

- The market for 5'-based methods is dominated by commercial solutions (specifically, 10x Genomics' 5' Gene Expression technology). While these commercial solutions provide streamlined workflows, the consumables and instrumentation required add substantial costs, ultimately limiting the number of samples that can be analyzed with a fixed budget. In contrast, a wide variety of academic solutions available for 3'-barcoded libraries (examples: Drop-Seq, Seq-Well, inDrops, Microwell-seq, SPLiT-seq)²⁰⁻²⁵ enable massively parallel scRNA-seq at a lower price point and provide greater customizability with regards to assay parameters.
- Importantly, the decision to perform BCR recovery and sequencing using our method can be determined after samples have been processed using 3'-barcoded platforms, allowing paired single-cell BCR and transcriptome data to be collected even if it was

not an initial goal of the experiment. As a result, we believe that our method effectively fills a niche in the single-cell technology space by broadening the number of technical solutions available to the single-cell immunologist.

To highlight these examples, we have performed the following revisions to our manuscript:

In the introduction: LINE 56

3`-barcoded libraries constructed using either academic (Drop-Seq, Seq-Well, inDrop, Microwell-seq, SPLiT-seq)^{20–25} or commercial (10x Genomics Single Cell 3` Gene Expression) platforms currently account for a substantial fraction of both previously published and newly-generated scRNA-seq data.

In the discussion: LINE 396

Our method is compatible with 3`-barcoded library produced with both Seq-Well and 10x Genomics, and we anticipate that it could be adapted to a broad range of single-cell library platforms, including emerging multi-modal approaches that aim to quantify surface epitope expression, chromatin accessibility, DNA methylation, and/or spatial location in combination with single-cell transcriptomes.

References

1. Dugan, H. L., Stamper, C. T., Li, L., Changrob, S., Asby, N. W., Halfmann, P. J., Zheng, N.-Y., Huang, M., Shaw, D. G., Cobb, M. S., Erickson, S. A., Guthmiller, J. J., Stovicek, O., Wang, J., Winkler, E. S., Madariaga, M. L., Shanmugarajah, K., Jansen, M. O., Amanat, F., Stewart, I., Utset, H. A., Huang, J., Nelson, C. A., Dai, Y.-N., Hall, P. D., Jedrzejczak, R. P., Joachimiak, A., Krammer, F., Diamond, M. S., Fremont, D. H., Kawaoka, Y. & Wilson, P. C. Profiling B cell immunodominance after SARS-CoV-2 infection reveals antibody evolution to non-neutralizing viral targets. *Immunity* **54**, 1290-1303.e7 (2021).
2. King, H. W., Orban, N., Riches, J. C., Clear, A. J., Warnes, G., Teichmann, S. A. & James, L. K. Single-cell analysis of human B cell maturation predicts how antibody class switching shapes selection dynamics. *Science immunology* **6**, (2021). PMID: 33579751
3. Zurbuchen, Y., Michler, J., Taeschler, P., Adamo, S., Cervia, C., Raeber, M. E., Acar, I. E., Nilsson, J., Warnatz, K., Soyka, M. B., Moor, A. E. & Boyman, O. Human memory B cells show plasticity and adopt multiple fates upon recall response to SARS-CoV-2. *Nat Immunol* 1–11 (2023). doi:10.1038/s41590-023-01497-y
4. Blundell, J. R. & Levy, S. F. Beyond genome sequencing: Lineage tracking with barcodes to study the dynamics of evolution, infection, and cancer. *Genomics* **104**, 417–430 (2014).
5. Gadala-Maria, D., Gidoni, M., Marquez, S., Vander Heiden, J. A., Kos, J. T., Watson, C. T., O'Connor, K. C., Yaari, G. & Kleinstein, S. H. Identification of Subject-Specific Immunoglobulin Alleles From Expressed Repertoire Sequencing Data. *Frontiers in Immunology* **10**, (2019).
6. Corcoran, M. M., Phad, G. E., Bernat, N. V., Stahl-Hennig, C., Sumida, N., Persson, M. A. A., Martin, M. & Hedestam, G. B. K. Production of individualized V gene databases reveals high levels of immunoglobulin genetic diversity. *Nat Commun* **7**, 13642 (2016). PMID: PMC5187446
7. Ralph, D. K. & Iv, F. A. M. Per-sample immunoglobulin germline inference from B cell receptor deep sequencing data. *PLOS Computational Biology* **15**, e1007133 (2019).
8. Vázquez Bernat, N., Corcoran, M., Hardt, U., Kaduk, M., Phad, G. E., Martin, M. & Karlsson Hedestam, G. B. High-Quality Library Preparation for NGS-Based Immunoglobulin Germline Gene Inference and Repertoire Expression Analysis. *Frontiers in Immunology* **10**, (2019).
9. Vázquez Bernat, N., Corcoran, M., Nowak, I., Kaduk, M., Castro Dopico, X., Narang, S., Maisonnasse, P., Dereuddre-Bosquet, N., Murrell, B. & Karlsson Hedestam, G. B. Rhesus and cynomolgus macaque immunoglobulin heavy-chain genotyping yields comprehensive databases of germline VDJ alleles. *Immunity* **0**, (2021).
10. Wu, X., Liu, Y., Jin, S., Wang, M., Jiao, Y., Yang, B., Lu, X., Ji, X., Fei, Y., Yang, H., Zhao, L., Chen, H., Zhang, Y., Li, H., Lipsky, P. E., Tsokos, G. C., Bai, F. & Zhang, X. Single-cell sequencing of immune cells from anticitrullinated peptide antibody positive and negative rheumatoid arthritis. *Nat Commun* **12**, 4977 (2021).
11. Cai, S., Chen, Y., Hu, Z., Zhou, T., Huang, Y., Lin, S., Gao, R., Zhong, J. & Dong, L. The landscape of T and B lymphocytes interaction and synergistic effects of Th1 and Th2 type response in the involved tissue of IgG4-RD revealed by single cell transcriptome analysis. *Journal of Autoimmunity* **133**, 102944 (2022).
12. Reily, C., Xu, N. & Crossman, D. K. Assigning immunoglobulin class from single-cell transcriptomes in IgA1-secreting versus membrane subpopulations. *Biotechniques* **70**, 89–99 PMID: PMC7983040
13. Stoeckius, M., Hafemeister, C., Stephenson, W., Houck-Loomis, B., Chattopadhyay, P. K., Swerdlow, H., Satija, R. & Smibert, P. Simultaneous epitope and transcriptome measurement in single cells. *Nature Methods* **14**, 865–868 (2017).
14. Chen, S., Lake, B. B. & Zhang, K. High-throughput sequencing of the transcriptome and chromatin accessibility in the same cell. *Nature Biotechnology* 2019 37:12 **37**, 1452–1457 (2019). PMID: 31611697

15. Cao, J., Cusanovich, D. A., Ramani, V., Aghamirzaie, D., Pliner, H. A., Hill, A. J., Daza, R. M., McFaline-Figueroa, J. L., Packer, J. S., Christiansen, L., Steemers, F. J., Adey, A. C., Trapnell, C. & Shendure, J. Joint profiling of chromatin accessibility and gene expression in thousands of single cells. *Science* **361**, 1380–1385 (2018). PMID: 30166440
16. Swanson, E., Lord, C., Reading, J., Heubeck, A. T., Genge, P. C., Thomson, Z., Weiss, M. D., Li, X.-J., Savage, A. K., Green, R. R., Torgerson, T. R., Bumol, T. F., Graybuck, L. T. & Skene, P. J. Simultaneous trimodal single-cell measurement of transcripts, epitopes, and chromatin accessibility using TEA-seq. *Elife* **10**, e63632 (2021). PMCID: PMC8034981
17. Stickels, R. R., Murray, E., Kumar, P., Li, J., Marshall, J. L., Di Bella, D. J., Arlotta, P., Macosko, E. Z. & Chen, F. Highly sensitive spatial transcriptomics at near-cellular resolution with Slide-seqV2. *Nat Biotechnol* **39**, 313–319 (2021).
18. Rodrigues, S. G., Stickels, R. R., Goeva, A., Martin, C. A., Murray, E., Vanderburg, C. R., Welch, J., Chen, L. M., Chen, F. & Macosko, E. Z. Slide-seq: A scalable technology for measuring genome-wide expression at high spatial resolution. *Science (New York, N.Y.)* **363**, 1463–1467 (2019). PMID: 30923225
19. Luo, C., Liu, H., Xie, F., Armand, E. J., Siletti, K., Bakken, T. E., Fang, R., Doyle, W. I., Stuart, T., Hodge, R. D., Hu, L., Wang, B.-A., Zhang, Z., Preissl, S., Lee, D.-S., Zhou, J., Niu, S.-Y., Castanon, R., Bartlett, A., Rivkin, A., Wang, X., Lucero, J., Nery, J. R., Davis, D. A., Mash, D. C., Satija, R., Dixon, J. R., Linnarsson, S., Lein, E., Behrens, M. M., Ren, B., Mukamel, E. A. & Ecker, J. R. Single nucleus multi-omics identifies human cortical cell regulatory genome diversity. *Cell Genomics* **2**, 100107 (2022).
20. Gierahn, T. M., Wadsworth, M. H., Hughes, T. K., Bryson, B. D., Butler, A., Satija, R., Fortune, S., Love, J. C. & Shalek, A. K. Seq-Well: portable, low-cost RNA sequencing of single cells at high throughput. *Nature Methods* **14**, 395–398 (2017).
21. Hughes, T. K., Wadsworth, M. H., Gierahn, T. M., Do, T., Weiss, D., Andrade, P. R., Ma, F., Silva, B. J. de A., Shao, S., Tsoi, L. C., Ordovas-Montanes, J., Gudjonsson, J. E., Modlin, R. L., Love, J. C. & Shalek, A. K. Highly efficient, massively-parallel single-cell RNA-Seq reveals cellular states and molecular features of human skin pathology. *bioRxiv* 689273 (2019). doi:10.1101/689273
22. Klein, A. M., Mazutis, L., Akartuna, I., Tallapragada, N., Veres, A., Li, V., Peshkin, L., Weitz, D. A. & Kirschner, M. W. Droplet barcoding for single-cell transcriptomics applied to embryonic stem cells. *Cell* **161**, 1187–1201 (2015). PMID: 26000487
23. Macosko, E. Z., Basu, A., Satija, R., Nemesh, J., Shekhar, K., Goldman, M., Tirosh, I., Bialas, A. R., Kamitaki, N., Martersteck, E. M., Trombetta, J. J., Weitz, D. A., Sanes, J. R., Shalek, A. K., Regev, A. & McCarroll, S. A. Highly parallel genome-wide expression profiling of individual cells using nanoliter droplets. *Cell* **161**, 1202–1214 (2015). PMID: 26000488
24. Chen, H., Liao, Y., Zhang, G., Sun, Z., Yang, L., Fang, X., Sun, H., Ma, L., Fu, Y., Li, J., Guo, Q., Han, X. & Guo, G. High-throughput Microwell-seq 2.0 profiles massively multiplexed chemical perturbation. *Cell Discov* **7**, 1–4 (2021).
25. Rosenberg, A. B., Roco, C. M., Muscat, R. A., Kuchina, A., Sample, P., Yao, Z., Gray, L., Peeler, D. J., Mukherjee, S., Chen, W., Pun, S. H., Sellers, D. L., Tasic, B. & Seelig, G. SPLiT-seq reveals cell types and lineages in the developing brain and spinal cord. *Science* **360**, 176–182 (2018). PMCID: PMC7643870

REVIEWERS' COMMENTS:

Reviewer #1 (Remarks to the Author):

Many thanks to the authors for seriously taking time to complete more experiments and perform further analysis to polish and enhance the contents! I am confident that the scientists in the field will enjoy reading and learn a lot from this manuscript!

Reviewer #2 (Remarks to the Author):

The authors have addressed all my concerns.

REVIEWERS' COMMENTS:

Reviewer #1 (Remarks to the Author):

Many thanks to the authors for seriously taking time to complete more experiments and perform further analysis to polish and enhance the contents! I am confident that the scientists in the field will enjoy reading and learn a lot from this manuscript!

We sincerely thank the reviewer for the validation.

Reviewer #2 (Remarks to the Author):

The authors have addressed all my concerns.

We thank the reviewer for reviewing our article.